# Structural basis for allosteric control of the SERCA-Phospholamban membrane complex by Ca[2+] and phosphorylation

Daniel K Weber[1], U Venkateswara Reddy[1], Songlin Wang[1], Erik K Larsen[2], Tata Gopinath[1], Martin B Gustavsson[1], Razvan L Cornea[1], David D Thomas[1], Alfonso De Simone[3,4], Gianluigi Veglia[1,2]*

[1]Department of Biochemistry, Molecular Biology and Biophysics, University of Minnesota, Minneapolis, United States; [2]Department of Chemistry, University of Minnesota, Minneapolis, United States; [3]Department of Life Sciences, Imperial College London, South Kensington, London, United Kingdom; [4]Department of Pharmacy, University of Naples 'Federico II', Naples, Italy

**Abstract** Phospholamban (PLN) is a mini-membrane protein that directly controls the cardiac Ca[2+]-transport response to β-adrenergic stimulation, thus modulating cardiac output during the fight-or-flight response. In the sarcoplasmic reticulum membrane, PLN binds to the sarco(endo) plasmic reticulum Ca[2+]-ATPase (SERCA), keeping this enzyme's function within a narrow physiological window. PLN phosphorylation by cAMP-dependent protein kinase A or increase in Ca[2+] concentration reverses the inhibitory effects through an unknown mechanism. Using oriented-sample solid-state NMR spectroscopy and replica-averaged NMR-restrained structural refinement, we reveal that phosphorylation of PLN's cytoplasmic regulatory domain signals the disruption of several inhibitory contacts at the transmembrane binding interface of the SERCA-PLN complex that are propagated to the enzyme's active site, augmenting Ca[2+] transport. Our findings address long-standing questions about SERCA regulation, epitomizing a signal transduction mechanism operated by posttranslationally modified bitopic membrane proteins.

*For correspondence:
vegli001@umn.edu

Competing interests: The authors declare that no competing interests exist.

## Introduction

Miniproteins are translated from small open-reading frames of 100–300 nucleotides in length and constitute a neglected portion of the human proteome (*Andrews and Rothnagel, 2014*). Most mini-proteins are membrane-embedded and act as regulators or ancillary proteins to enzymes or receptors (*Anderson et al., 2016*; *Ma et al., 2014*; *Slavoff et al., 2013*). Among the most critical miniproteins is phospholamban (PLN), a bitopic membrane polypeptide that regulates the function of the sarco(endo)plasmic reticulum Ca[2+]-ATPase (SERCA) in cardiac muscle (*Tada et al., 1979*). PLN directly controls cardiac output by maintaining SERCA's activity within a tight physiological window (*Bers, 2002*). SERCA is a 10-transmembrane (TM) helices pump that promotes diastole by removing Ca[2+] from the sarcoplasm and restoring high Ca[2+] concentrations in the sarcoplasmic reticulum (SR) in preparation for the next systole (*Bers, 2002*). As with other P-type ATPases, SERCA is fueled by ATP and cycles between two major conformational states *E*1 and *E*2, of high and low Ca[2+]-affinity, respectively (*Dyla et al., 2020*). In cardiomyocytes, PLN is expressed in fourfold molar excess of SERCA, suggesting that this endogenous regulator is permanently bound to the enzyme in a 1:1 stoichiometric ratio (*Ferrington et al., 2002*). PLN binds the ATPase via intramembrane pro-tein-protein interactions, lowering its apparent Ca[2+] affinity and stabilizing the *E*2 state of the pump (*Bers, 2002*; *MacLennan and Kranias, 2003*). SERCA/PLN inhibitory interactions are relieved upon β-adrenergic stimulation, which unleashes cAMP-dependent protein kinase A (PKA) to phosphorylate

PLN's cytoplasmic domain at Ser16, enhancing $Ca^{2+}$ transport by SERCA and augmenting heart muscle contractility (*Chu et al., 2000*). Ablation, point mutations, or truncations of PLN have been linked to congenital heart disease (*MacLennan and Kranias, 2003*). Despite multiple crystal structures of SERCA alone (*Dyla et al., 2020*) and several structural studies of PLN free and bound to SERCA (*Segrest et al., 1990*; *James et al., 2012*; *Karim et al., 2006*), the inhibitory mechanism of PLN and its reversal upon phosphorylation or $Ca^{2+}$ increase are still unknown. Mutagenesis data and molecular modeling suggest that SERCA regulation occurs through electrostatic and hydrophobic interactions between the helical TM region of PLN and the binding groove of the ATPase formed by helices TM2, TM6, and TM9 (*Toyoshima et al., 2003*). Upon phosphorylation, or binding SERCA, however, the helical TM domain of PLN does not undergo significant changes in secondary structure (*Seidel et al., 2008*; *Akin et al., 2013*; *Gustavsson et al., 2013*). As a result, X-ray crystallography (*Akin et al., 2013*) and other structural techniques (e.g. EPR or NMR) have offered incomplete mechanistic insights into the regulatory process.

Here, we reveal the elusive signal transduction mechanism responsible for phosphorylation-induced activation of the SERCA/PLN complex using a combination of oriented-sample solid-state NMR (OS-ssNMR) spectroscopy and dynamic structural refinement by replica-averaged orientational-restrained molecular dynamics simulations (RAOR-MD) (*De Simone et al., 2014*; *Sanz-Hernández et al., 2016*). The analysis of anisotropic $^{15}N$ chemical shifts (CSs) and $^{15}N$-$^1H$ dipolar couplings (DCs) of PLN alone and in complex with SERCA in magnetically aligned lipid bicelles unveiled collective topological changes of PLN's inhibitory TM domain in response to Ser16 phosphorylation. Specifically, the local perturbations of phosphorylation were allosterically transmitted via an order-disorder transition of the juxtamembrane helical residues involved in several inhibitory interactions with SERCA. This intramembrane regulatory mechanism represents a potential paradigm of the structural basis of SERCA activity modulation by other regulins (e.g. sarcolipin, myoregulin, DWORF, etc.) (*Anderson et al., 2016*) in response to different physiological cues.

## Results

### The TM domain of PLN undergoes a topological two-state equilibrium

In lipid membranes, PLN adopts an *L*-shaped conformation, with a membrane-adsorbed, amphipathic regulatory region (domain Ia, M1 to T17) connected by a short loop (Ile18 to Gln22) to a helical inhibitory region (domains Ib, Gln23 to Asn30; and domain II, Leu31 to Leu52), which crosses the SR membrane (*Traaseth et al., 2009*; *Verardi et al., 2011*). In its storage form, PLN is pentameric (*Verardi et al., 2011*; *Vostrikov et al., 2013*; *Mravic et al., 2019*) and de-oligomerizes into active L-shape monomers (*Traaseth et al., 2009*). The dynamic cytoplasmic region undergoes an order-disorder transition between tense (T) and relaxed (R) states, with the latter promoted by Ser16 phosphorylation (*Seidel et al., 2008*; *Gustavsson et al., 2013*; *Gustavsson et al., 2012*). Upon binding SERCA, domain Ia transitions to a more rigid and non-inhibitory bound (B) state, becoming more populated upon phosphorylation (*Seidel et al., 2008*; *Gustavsson et al., 2013*). How does Ser16 phosphorylation signal the reversal of inhibition to the TM region? Since the inhibitory TM region is ~45 Å away from Ser16 and ~20 Å from SERCA's $Ca^{2+}$-binding sites, we speculated that both phosphorylation (of PLN) and $Ca^{2+}$ binding (to SERCA) must transmit conformational and topological changes across the membrane, thus allosterically modulating SERCA's function.

Residue-specific anisotropic NMR parameters such as CSs and DCs are exquisitely suited to describe topological transitions such as tilt, bend, and torque of TM proteins in lipid bilayers near-physiological conditions (*Opella and Marassi, 2004*; *Das et al., 2013*). Their analysis by OS-ssNMR requires that membrane-embedded proteins are uniformly oriented relative to the static magnetic field ($\mathbf{B_0}$). Therefore, we reconstituted PLN free and in complex with SERCA into magnetically aligned lipid bicelles (*Sanders and Landis, 1995*). Since monomeric PLN is the functional form (*Zvaritch et al., 2000*), we utilized a cysteine-null monomeric mutant of PLN (Cys36Ala, Cys41Phe, and Cys46Ala) (*Karim et al., 2000*) exclusively throughout this study. Both unphosphorylated and phosphorylated (pPLN) variants of PLN were expressed recombinantly, while SERCA was purified from mammalian tissues (*Stokes and Green, 1990*). Since lipid bicelles orient spontaneously with the normal of the membrane ($\vec{n}$) perpendicular to $\mathbf{B_0}$, we doped the sample with $Yb^{3+}$ ions to change the magnetic susceptibility and orient the lipid membranes with $\vec{n}$ parallel to $\mathbf{B_0}$. This expedient

doubles the values of CSs and DCs and increases the resolution of the NMR spectra (*Prosser et al., 1996*; *Prosser et al., 1998*). *Figure 1A* shows the 2D [$^{15}$N-$^{1}$H] sensitivity-enhanced (SE)-SAMPI4 (*Gopinath and Veglia, 2009*) separated local field (SLF) spectra of free PLN and pPLN. Due to PLN's intrinsic conformational dynamics, the SLF spectra visualize only its TM region. The spectra display the typical wheel-like pattern diagnostic of a helical conformation for both the TM domains of PLN and pPLN. Residue-specific assignments were carried out on free PLN using a combination of a 3D SE-SAMPI4-proton-driven spin diffusion (PDSD) spectrum (*Mote et al., 2011*), selective $^{15}$N-labeled samples, and predictions from MD simulations (*Weber and Veglia, 2020*; *Table 1*; *Figure 1—figure supplements 1–3*). To obtain PLN's topology in lipid bilayer, the assigned resonances were fit to idealized *P*olar *I*ndex *S*lant *A*ngle (PISA) models extracting whole-body tilt (θ) and rotation or azimuthal (ρ) angles (*Denny et al., 2001*; *Marassi and Opella, 2000*), which for free PLN was θ = 37.5 ± 0.7° and $\rho_{L31}$ = 201 ± 4°, and for pPLN θ = 34.8 ± 0.5° and $\rho_{L31}$ = 201 ± 4°, where $\rho_{L31}$ is the rotation angle referenced to Leu31. Notably, the high resolution of the oriented SLF spectra of PLN show two distinct sets of peaks (*Figure 1B*), with populations unevenly distributed. The average population of the minor state estimated from the normalized peak intensities is approximately 30 ± 9 %. Remarkably, the resonances of the minor population overlap almost entirely with those of pPLN (*Figure 1B*), revealing a topological equilibrium in which the TM region of PLN interconverts between two energetically different orientations. We previously showed that PLN phosphorylation shifts the conformational equilibrium toward the *R* state (*Gustavsson et al., 2013*), releasing the interactions with the lipid membranes of domain Ia (*Figure 1C*). Our OS-ssNMR data show that these phosphorylation-induced effects propagate to the TM domains, shifting the topological equilibrium toward the less populated state.

## PLN phosphorylation by PKA signals a rearrangement of the SERCA/PLN-binding interface

To investigate how Ser16 phosphorylation allosterically affects the inhibitory TM binding interface, we reconstituted the SERCA/PLN complex in lipid bicelles and studied it by OS-ssNMR. The alignment of mammalian SERCA in bicelles was confirmed by cross-linking the most reactive cysteines with a trifluoromethylbenzyl (TFMB)-methanethiosulfonate (MTS) tag to probe its alignment by $^{19}$F NMR (*Figure 1D*; *Figure 1—figure supplement 4*). Five of the 24 cysteines of SERCA were uniquely labeled as monitored by solution NMR in isotropic bicelles (q = 0.5), with two positions being the most prominent. In anisotropic bicelles (q = 4.0) and at low temperature, the ssNMR spectrum of $^{19}$F-SERCA consists of a single unresolved $^{19}$F resonance due to the rapid reorientation of the enzyme in the isotropic phase. Upon increasing the temperature, the $^{19}$F-SERCA/bicelle complex orients with $\vec{n}$ perpendicular to **B**$_0$, and the $^{19}$F resonance becomes anisotropic as a triplet with 1.3 kHz dipolar coupling and no indication of any powder pattern associated with non-aligned SERCA (*Glaser et al., 2004*). *Figure 1E* shows the 2D SLF spectra of PLN and pPLN in complex with SERCA. To maintain a functional and stable complex, we used a lipid-to-complex molar ratio of 2000:1, equivalent to one SERCA per bicelle at this q-ratio (*Glover et al., 2001*), with PLN concentration 10 times less than in the SERCA-free samples. Therefore, the signal-to-noise ratio in the oriented spectra is significantly reduced relative to the free forms. Nonetheless, the SLF spectra of both SERCA/PLN and SERCA/pPLN complexes show the wheel-like patterns typical of the α-helical domains with selective exchange broadening for resonances located at the protein-protein binding interface (*Figure 1F*). The assigned peaks associated with the helical domain II were fit to the ideal PISA model, yielding θ = 33.2 ± 1.2° and $\rho_{L31}$ = 193 ± 7°. Therefore, upon binding SERCA, PLN requires a distinguishable -4.3 ± 1.4° change in tilt (p = 4.3 × 10$^{-15}$) and a less significant -8 ± 8° change in rotation (p = 0.00027). These error bounds factor the linewidths and variation associated with substituting ambiguous assignments into the PISA fitting (parentheses of *Figure 1E*). Similarly, the PISA model for pPLN was fit to θ = 30.4 ± 1.1° and $\rho_{L31}$ of 197 ± 4°, suggesting that the topology of the TM domain requires adjustments of -3.4 ± 1.2° (p = 2.9 × 10$^{-14}$) and -4 ± 6° (p = 0.0013) to form a complex with the ATPase.

Reductions in the TM helix tilt angle, which accompanied phosphorylation and complex formation, also coincided with a dramatic broadening of peaks in the cluster of isotropic resonances around 140 ppm (*Figure 1E*). These resonances are attributed to the dynamic domain Ib residues, and their disappearance is consistent with SERCA binding, which requires the unwinding of the

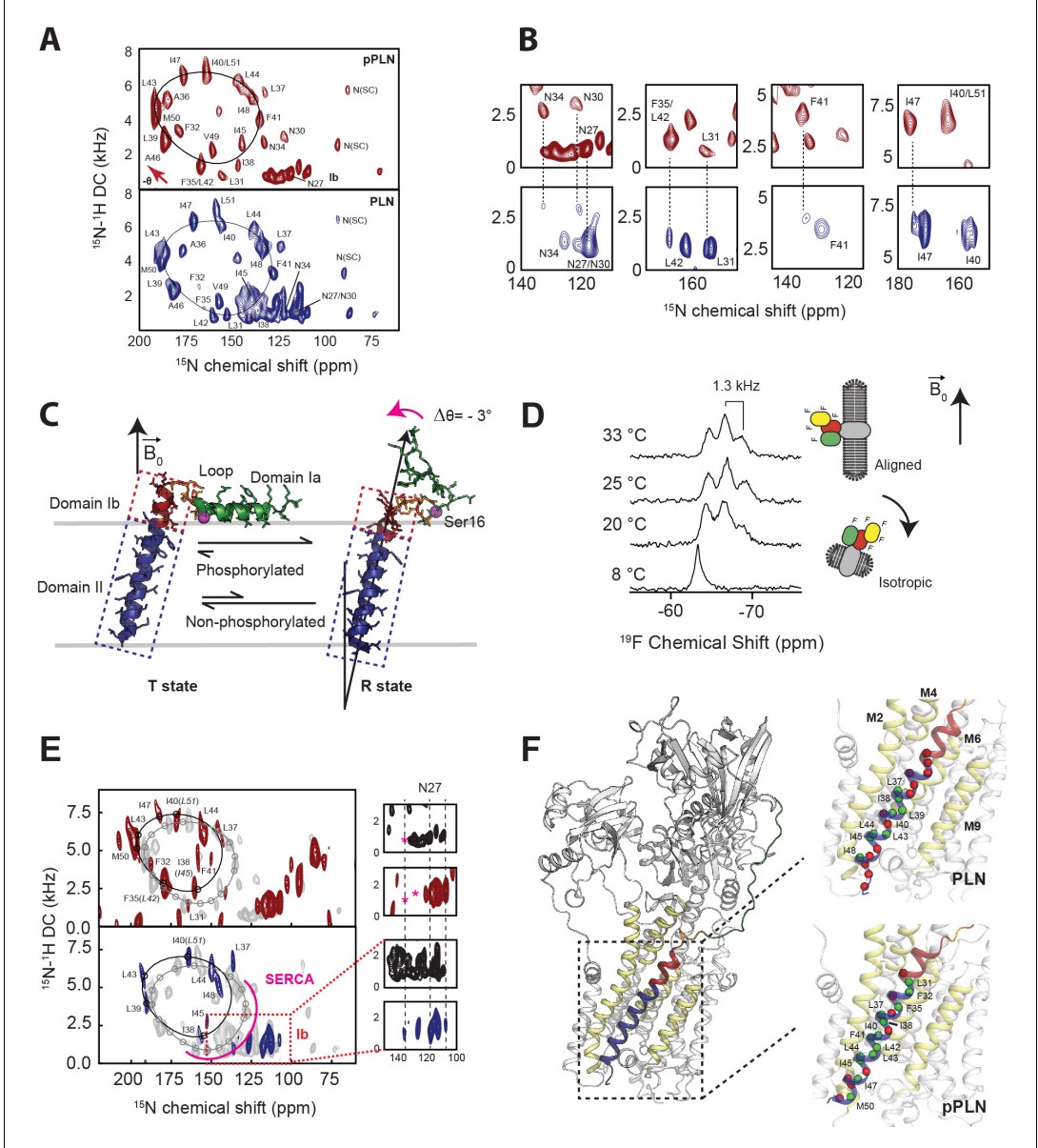

**Figure 1.** Topological equilibrium of PLN and pPLN free and bound to SERCA in lipid bilayers detected by OS-ssNMR. (**A**) 2D [$^{15}$N-$^1$H] SE-SAMPI4 spectra of PLN and pPLN reconstituted into aligned lipid bicelles. The fitting of resonance patterns with PISA wheels for an ideal helix [($\phi$, $\psi$)=(−63°, −42°)] is superimposed. (**B**) Expanded regions of PLN $^{15}$N-labeled at N, L, F, or I residues (lower panel, blue contours) showing two populations. The upper panels (red) are the corresponding regions for the U-$^{15}$N labeled pPLN. U-$^{15}$N labeled spectra were acquired at higher signal-to-noise to observe the second population. (**C**) Structures of the *T* (PDB 2KB7 *Traaseth et al., 2009*) and *R* (PDB 2LPF *De Simone et al., 2013*) states for PLN. (**D**) $^{19}$F NMR spectra of TFMB-tagged SERCA reconstituted into anisotropic (*q* = 4) bicelles at variable temperatures. (**E**) 2D [$^{15}$N-$^1$H] SE-SAMPI4 spectra of uniformly $^{15}$N labeled PLN (blue, lower panel) and pPLN (red, upper panel) bound to SERCA in the absence of Ca$^{2+}$ (*E2* state). Spectra are overlaid with PLN or pPLN in their free forms (gray). PISA wheels are overlaid, showing assigned residues (black points) used to fit helical tilt and rotation angles. Ambiguous assignments are shown in parentheses. The region corresponding to domain Ib is expanded to show peak broadening (asterisk) following the transition of PLN's cytoplasmic region to the *B* state. (**F**) Selected structure of the SERCA/PLN complex. Expanded region shows visible (green spheres, labeled) and broadened (red spheres) residues mapped onto domain II of PLN and pPLN in complex with SERCA.

The online version of this article includes the following figure supplement(s) for figure 1:

**Figure supplement 1.** Assignment by 3D OS-ssNMR.

**Figure supplement 2.** Assignment of OS-ssNMR spectra by selective labeling and un-labeling experiments.

**Figure supplement 3.** Prediction of assignments from unrestrained MD simulation of truncated PLN.

**Figure supplement 4.** Synthesis and oriented NMR of the trifluoromethylbenzyl (TFMB)-methanethiosulfonate (MTS) tag.

**Table 1.** OS-ssNMR assignments.

Summarized $^{15}$N CS (ppm) and $^{15}$N-$^{1}$H DC (kHz) assignments of PLN and pPLN reconstituted alone in bicelles or in complex with SERCA in either $E$2 (Ca$^{2+}$-free) or $E$1 (5 mM Ca$^{2+}$) states.

| Res. | PLN | | pSer16-PLN | | PLN + SERCA ($E$2) | | pSer16-PLN +SERCA ($E$2) | | PLN + SERCA ($E$1) | | pSer16-PLN +SERCA ($E$1) | |
|---|---|---|---|---|---|---|---|---|---|---|---|---|
| | CS | DC | CS | DC | CS | DC | CS | DC | CS | DC | CS | DC |
| N27 | 117.6* | 1.19* | 118.1 | 0.90 | - | - | - | - | - | - | - | - |
| N30 | 117.6* | 1.19* | 121.5 | 3.10 | - | - | - | - | - | - | - | - |
| L31 | 152.8 | 0.94 | 155.2 | 0.74 | - | - | 164.3 | 1.26 | - | - | - | - |
| F32 | 167.7 | 2.46 | 177.9 | 3.38 | - | - | 187.6 | 4.07 | 181.6 | 3.78 | 186.6 | 4.56 |
| N34 | 125.8 | 1.36 | 132.4 | 2.71 | - | - | - | - | - | - | - | - |
| F35 | 164.7 | 1.17 | 166.2* | 1.36* | - | - | 179.1 | 2.86 | 176.4 | 2.30 | - | - |
| A36 | 176.5 | 4.74 | 184.3 | 5.28 | 186.5 | 5.72 | 191.0 | 6.37 | 190.4 | 5.91 | 192.8 | 6.76 |
| L37 | 123.9 | 4.89 | 132.2 | 5.71 | 136.5 | 6.66 | 146.0 | 6.57 | 146.1 | 4.73 | 151.7 | 5.87 |
| I38 | 140.8 | 1.03 | 146.5 | 1.40 | 156.9 | 1.86 | 160.3 | 2.54 | 156.4 | 2.11 | 164.9 | 2.14 |
| L39 | 181.7* | 2.31* | 185.7* | 2.86* | 190.3 | 3.88 | 194.4 | 3.99 | 194.5 | 3.76 | 198.5 | 4.75 |
| I40 | 155.7 | 6.19 | 163.5* | 6.88* | 164.0 | 7.04 | 171.5 | 6.87 | 166.7 | 6.99 | 169.9 | 7.25 |
| F41 | 127.8 | 3.39 | 134.6 | 4.19 | - | - | 150.0 | 4.46 | - | - | 148.4 | 4.33 |
| L42 | 160.5 | 0.91 | 166.2* | 1.36* | 175.5 | 1.74 | 179.1 | 2.86 | - | - | 181.1 | 2.63 |
| L43 | 187.5* | 4.17* | 190.9* | 4.46* | 192.8 | 5.77 | 198.3 | 5.76 | 196.5 | 5.89 | - | - |
| L44 | 138.0 | 6.02 | 145.6 | 6.19 | 149.9 | 6.22 | 155.5 | 6.26 | 156.0 | 5.19 | 155.1 | 6.65 |
| I45 | 139.2 | 1.80 | 144.5 | 2.64 | 152.7 | 2.74 | 160.3 | 2.54 | - | - | 162.1 | 3.57 |
| A46 | 181.7* | 2.31* | 185.7* | 2.86* | - | - | 191.4 | 2.84 | - | - | - | - |
| I47 | 170.5 | 6.35 | 175.9 | 6.65 | - | - | 181.8 | 7.23 | 179.5 | 6.74 | 181.9 | 7.99 |
| I48 | 132.7 | 4.88 | 139.2 | 5.39 | 145.5 | 5.16 | 154.3 | 6.03 | 151.2 | 4.94 | - | - |
| V49 | 157.1 | 1.75 | 160.1 | 2.26 | - | - | - | - | - | - | - | - |
| M50 | 187.5* | 4.17* | 190.9* | 4.46* | - | - | 198.0 | 5.14 | 202.5 | 4.80 | 199.9 | 5.94 |
| L51 | 158.3 | 7.06 | 163.5* | 6.88* | - | - | - | - | - | - | - | - |

*These peaks are overlapped.

juxtamembrane region, and a concomitant reduction of the tilt angle to re-establish hydrophobic matching with the thickness of the lipid bilayer (*Karim et al., 2006*; *Gustavsson et al., 2013*). Therefore, for free and bound PLN, we find that communication between cytoplasmic and intramembrane environments is transduced via domain Ib dynamics. Analysis of the SLF spectra also shows that phosphorylation of PLN at Ser16 restores the intensities of most resonances except for those at the upper binding interface (i.e. Asn30, Leu31, Asn34, and Phe41). These spectral changes suggest a reorganization of PLN-SERCA packing interactions, rather than a complete dissociation of the complex, consistent with prior MAS-ssNMR, EPR, and FRET measurements (*Karim et al., 2006*; *Gustavsson et al., 2013*; *Dong and Thomas, 2014*; *Bidwell et al., 2011*).

## Dynamic structural refinement of the SERCA-PLN complexes

To better understand how PLN phosphorylation relieves SERCA inhibition, we determined the structural ensembles of the SERCA/PLN and SERCA/pPLN complexes in the non-inhibitory bound (*B*) state by incorporating the data from our experimental measurements into RAOR-MD samplings (*De Simone et al., 2014*; *Sanz-Hernández et al., 2016*). This dynamic refinement methodology employs full atomic MD simulations in explicit lipid membranes and water and utilizes restraints from sparse datasets to generate experimentally driven structural ensembles. As starting coordinates for our samplings, we used the X-ray structure of *E*2-SERCA/PLN, where a super-inhibitory mutant of PLN was used to stabilize the complex for crystallization (*Akin et al., 2013*). We docked the TM domains of PLN using restraints obtained from chemical cross-linking experiments for both

cytoplasmic and luminal sites (*Toyoshima et al., 2003*; *Chen et al., 2006*; *Chen et al., 2003*; *Figure 2—figure supplement 1A*). The dynamic cytoplasmic region (loop and domain Ia), which was not resolved in the crystal structure, was constrained to a region between the nucleotide-binding (N) and phosphorylation (P) domains of SERCA via paramagnetic relaxation enhancements (PRE) obtained from MAS-ssNMR of the complex in the *B*-state (*Gustavsson et al., 2013*). Chemical shifts have indicated that PLN's domain Ia binds SERCA's cytoplasmic headpiece in an extended conformation (*Gustavsson et al., 2013*). Intermolecular PREs included quenching of PLN domain Ia resonances by SERCA bearing a paramagnetic (1-oxyl-2,2,5,5-tetramethylpyrroline-3-methyl) methanethiosulfonate spin label (MTSSL) on Cys674 and quenching of $^{13}C$-methyl thiocysteines (MTC), engineered throughout SERCA's N and P domains, by MTSSL-labeled PLN (*Figure 2—figure supplement 1A*). Due to the dynamic nature of domain Ia and the ambiguity of $^{13}C$-MTC assignments, all PRE restraints were implemented as 30 Å boundary conditions to confine the conformational freedom of PLN domain Ia to the approximate binding site (*Figure 2—figure supplement 1B*). Using this scheme, the overall profile of the average pairwise distances for residues in domain Ia and loop to the spin-label at Cys674 matches the PRE measurements, with the minimal distance (i.e. maximum PRE effect) observed for PLN-Tyr6 (*Figure 2—figure supplement 1C,D*).

CSs and DCs from OS-ssNMR were applied to the TM region as ensemble-averaged restraints across eight replicas. Back-calculated CS and DC values for PLN and pPLN were in excellent agreement with experiments (*Figure 2—figure supplement 1E*). Average back-calculated tilt angles of 32.8° and 30.4° for PLN and pPLN, respectively, matched PISA fits to experimental values (*Figure 2—figure supplements 1F* and *2*). All pairwise distances between previously reported cross-linkable positions (*Toyoshima et al., 2003*; *Chen et al., 2006*; *Chen et al., 2003*; *Jones et al., 2002*) were distributed within acceptable ranges (*Figure 2—figure supplement 1G*). Convergence of Q-factors of DC and DC restraints during the annealing cycles is shown in *Figure 2—figure supplement 3A*. Although not used as an initial docking restraint, cytoplasmic residues PLN Lys3 and SERCA Lys397 were also partially distributed within a distance consistent with previously reported cross-linking (*James et al., 1989*). Overall, the resulting structural ensembles were in excellent agreement with all the available experimental data for both complexes.

While TM and cytoplasmic regions of PLN were restrained throughout the simulations, SERCA was unrestrained and displayed a significant conformational heterogeneity over the timescale sampled. To assess SERCA's conformational landscapes, we used principal component analysis (PCA), clustering the ensembles according to two key motions, exemplifying a combined opening of the cytoplasmic headpiece involving a hinge-like displacement of the N domain and rotation of the A domain away from the P domain (PC1) and planar rotations separating the N and A domains (PC2) (*Figure 2A,B*; *Figure 2—figure supplement 4*, *Figure 2—videos 1*, *2*). These motions differentiate the *E*1 and *E*2 states of SERCA, as shown by the projections of crystal structures onto the PCA map (*Figure 2C,D*). The SERCA/PLN complex spans four distinct clusters, while the SERCA/pPLN complex spans eight (see *Figure 2—figure supplement 5* for representative structures). Both ensembles converged with respect to both PC1 and PC2 motions based on the Kullback-Leibler divergence (KLD) plot of the 2D PCA histograms. Note that the convergence of the SERCA/pPLN ensemble was slower due to the additional conformational dynamics (*Figure 2—figure supplement 3B*). When bound to PLN, SERCA mostly retains the compact *E*1-like headpiece present in the crystal structure and predominantly occupied a highly compact cluster (3) resembling nucleotide-bound *E*1 states and a cluster (1) intermediate toward the *E*2 states. Albeit biased along the *E*1 coordinate of PC2, similar states were present for pPLN, but the interaction of pSer16 with Arg604 weakens the Asp601-Thr357 and Arg604-Leu356 hydrogen bonds at the hinge of the N and P domains, leading to four additional open states (clusters 5–8). Separate clusters correspond to successive breakages of interdomain hydrogen bonds in the headpiece. Salt bridges between PLN-Ser16 to SERCA-Arg460 and PLN-Arg14 to SERCA-Glu392 were also found to stabilize these additional open states (*Figure 2—figure supplement 6*). These open states resemble off-pathway crystal structures solved for the $Ca_2E$1 state observed in the absence of nucleotide (*Dyla et al., 2020*; *Toyoshima et al., 2000*; *Jensen et al., 2006*). For all clusters, the binding interactions near the Ser16 position were more persistent for pPLN than PLN (*Figure 2E,F*; *Figure 2—figure supplement 5*). Although our experimental data cannot verify these conformational landscapes of SERCA, our ensembles nonetheless strongly suggest that phosphorylation of PLN has long-range effects on the complex well beyond the protein-protein binding interface.

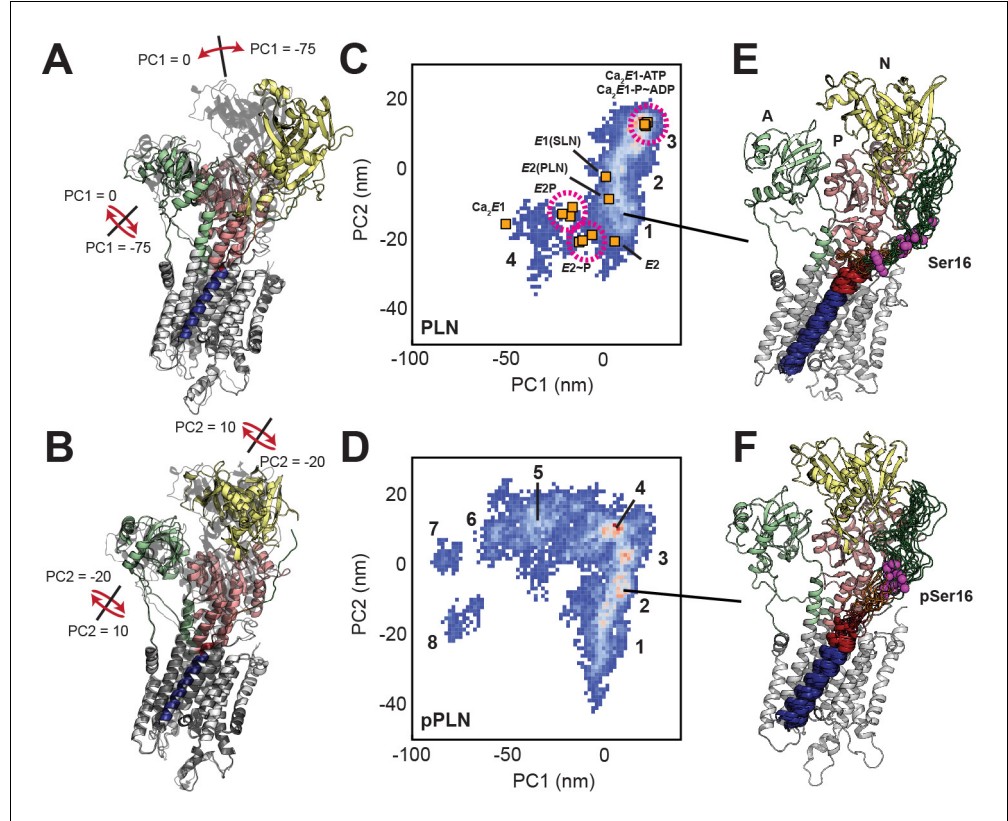

**Figure 2.** Conformational ensembles and energy landscapes of the SERCA/PLN and SERCA/pPLN complexes. (**A**, **B**) Depiction of headpiece movements associated with the first (**A**) and second (**B**) principal components. Structures with the highest PC values are shown as transparent black. (**C**, **D**) PCA histograms of SERCA/PLN (**C**) and SERCA/pPLN (**D**) structural ensembles with projections of crystal structures in various states: $Ca_2E1$-ATP (***Toyoshima and Mizutani, 2004***; ***Sørensen et al., 2004***), $Ca_2E1 \sim$ P-ADP (***Sørensen et al., 2004***; ***Olesen et al., 2007***; ***Toyoshima et al., 2004***), $Ca_2E1$ (***Toyoshima et al., 2000***), $E1$-SLN (***Winther et al., 2013***), $E2$-PLN (***Akin et al., 2013***), $E2$P (***Olesen et al., 2007***; ***Toyoshima et al., 2007***), $E2\sim$P (***Olesen et al., 2007***; ***Toyoshima et al., 2007***; ***Bublitz et al., 2013***), and $E2$ (***Toyoshima and Nomura, 2002***). Clusters are numbered. (**E**, **F**) Top 20 most representative structures of PLN (**E**, cluster 1) and pPLN (**F**, cluster 2) bound to SERCA from the most representative state.

The online version of this article includes the following video and figure supplement(s) for figure 2:

**Figure supplement 1.** Restraints for RAOR-MD structural refinement.

**Figure supplement 2.** Experimental and back-calculated RAOR-MD SLF spectra of the PLN/SERCA complex.

**Figure supplement 3.** Convergence of OS-ssNMR restraints and the conformational landscape of SERCA.

**Figure supplement 4.** Headpiece dynamics of SERCA in RAOR-MD.

**Figure supplement 5.** RAOR-MD clustering.

**Figure supplement 6.** Example open state snapshot (PCA cluster 6) of SERCA stabilized by electrostatic interactions between pSer16 and the N domain.

**Figure 2—video 1.** Movie of the SERCA/PLN complex motion along PC1.

https://elifesciences.org/articles/66226#fig2video1

**Figure 2—video 2.** Movie of the SERCA/PLN complex motion along PC2.

https://elifesciences.org/articles/66226#fig2video2

From the analysis of the structural ensembles of the two complexes, it emerges that the relief of inhibition (i.e. activation) occurs via a rearrangement of the intramembrane contacts between the TM region of pPLN and SERCA, with a reconfiguration of electrostatic interactions near the phosphorylation site and a disruption of packing at the protein-protein interface (***Figure 3A,B***). The interactions between the cytoplasmic regions are transient and highly dynamic, resembling the conformational ensembles of intrinsically disordered complexes (***Olivieri et al., 2020***). For both

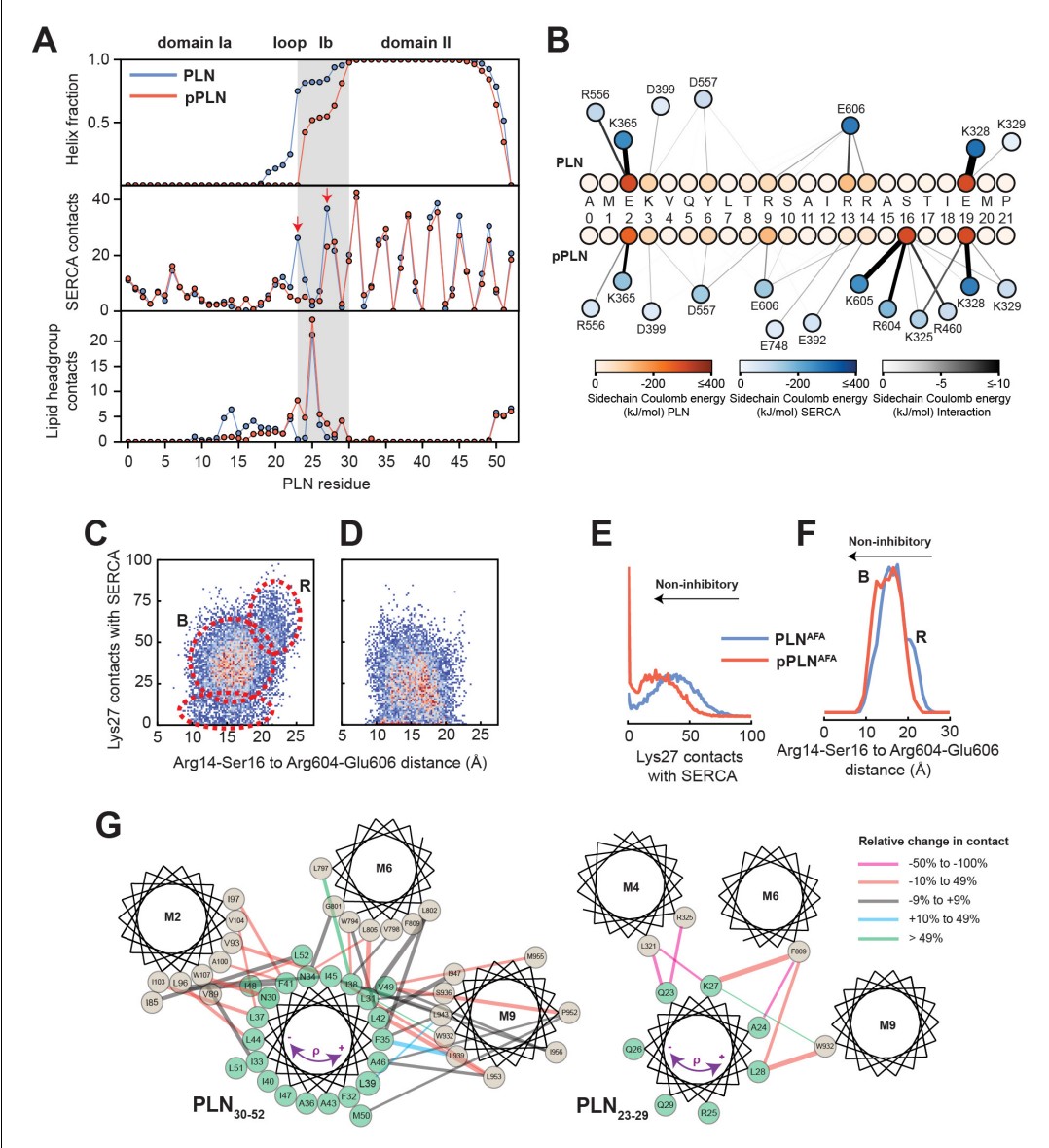

**Figure 3.** Mechanism for reversal of PLN inhibition by phosphorylation. (A) Ensemble-averaged per-residue (PLN) structural analysis (upper panel), intermolecular SERCA contacts (middle panel), and intermolecular lipid headgroup contacts (lower panel). A contact is defined when any PLN atom comes within 3.5 Å of any SERCA or lipid headgroup (i.e. part of the phosphate or choline moiety) atom for any given frame. (B) Spider plot of pairwise electrostatic interactions between cytoplasmic residues and SERCA in RAOR-MD conformational ensembles. (C, D) 2D Histograms correlating the distances between the cytoplasmic binding interfaces, defined by the center of masses of Arg14 to Ser16 of PLN and Arg604 to Glu606 of SERCA, to the inhibitory intermolecular contacts of PLN Lys27 for the SERCA/PLN (C) and SERCA/pPLN (D) ensembles. (E, F) Corresponding 1D histograms for Lys27 contacts (E) and binding of the cytoplasmic domain (F). (G) Disruption of the inhibitory TM pairwise interactions detected in the RAOR-MD conformational ensembles. Linewidths of the interhelical contacts are scaled to average contacts per frame for the non-phosphorylated complex and colored by the relative change observed with phosphorylation. Directions of the purple arrows exemplify clockwise or counterclockwise rotations of the TM domain of PLN during the trajectories.

The online version of this article includes the following figure supplement(s) for figure 3:

**Figure supplement 1.** Summary of inhibitory TM contacts.

complexes, we observe persistent interactions between PLN-Glu2 and SERCA-Lys365, PLN-Glu19, and SERCA-Lys328, and to a lesser extent PLN-Lys3 and SERCA-Asp399 and PLN-Tyr6 and SERCA-Asp557. For PLN, however, the arginine residues (Arg9, Arg13, and Arg14) interact transiently with SERCA-Glu606, while for pPLN, the phosphate group at Ser16 interacts strongly with SERCA-

Arg604 and SERCA-Lys605. Residual interactions observed for PLN residues Arg13, Arg14, Ser16, and Thr17 with lipid headgroups were abolished with pPLN (*Figure 3A*), consistent with the correlation between membrane-detachment of domain Ia and reduced inhibition (*Gustavsson et al., 2012*; *Gustavsson et al., 2011*). Interestingly, we detected the formation of intramolecular salt-bridges between the phosphate of Ser16 and PLN-Arg9, PLN-Arg13, and PLN-Arg14, causing domain Ia to adopt a compact conformation as previously suggested by fluorescence data (*Li et al., 2003*; *Li et al., 2004*; *Tables 2* and *3*). These cytoplasmic protein-protein interactions destabilized PLN's domain Ib and consequently severed inhibitory intermolecular contacts with SERCA's TM helices involving Gln23-Leu321/Arg325 (M4), Lys27-Phe809 (M6), and Asn30-Trp107 (M2), while the intermolecular contacts involving domain II are mostly retained (*Figure 3A* and *Figure 3—figure supplement 1*). For pPLN, the disrupted domain Ib-SERCA interactions were substituted by the interactions with the lipid headgroup, suggesting that the membrane itself may also play a localized role in modulating inhibition (*Figure 3A*). In fact, hydrophobic substitutions within domain Ib have been identified as hotspots for engineering super-inhibitory PLN mutants (i.e. Asn/Lys27Ala and Asn30Cys), which exhibit stable helical structure well into the loop domain (*Akin et al., 2013*; *Kimura et al., 1998*; *Akin et al., 2010*). Importantly, these structural ensembles capture the order-disorder dynamics of domain Ib resonances observed in the OS-ssNMR spectra and suggested by previous NMR and EPR studies (*Gustavsson et al., 2011*). Destabilization and detachment of this region is consistent with the reappearance of exchange-broadened interfacial resonances of domain II paralleled by broadening of the resonances of the dynamic domain Ib in the SLF spectra of the SERCA/pPLN complex (*Figure 1E*). The interactions of domain Ia (Arg13-Ser16) and inhibitory contacts of domain Ib (at Lys27) with SERCA appear to be mutually exclusive (*Figure 3C–F*). In both complexes, the electrostatic interactions of PLN-Arg13, PLN-Arg14, or PLN-pSer16 with SERCA's Arg604-Glu606 stretch cause the detachment of PLN's domain Ib and the consequent weakening of the inhibitory interaction (*Figure 3C–G*). This illustrates the regulatory role of the *B* state of PLN for relieving inhibition and the super-inhibitory activity of domain Ia-truncated PLN (*Gustavsson et al., 2013*).

## Phosphorylation disrupts correlated motions between PLN and SERCA's $Ca^{2+}$-binding sites

To assess the effects of PLN's phosphorylation on $Ca^{2+}$ transport, we calculated the topological correlations of the TM's tilt angle fluctuations between PLN's and SERCA's TM helices (*Figure 4A–D*). When PLN is bound to SERCA, we observe a dense network of correlated motions between PLN's TM region and the binding groove (TM2, TM6, and TM9), as well as a dense cluster of correlations involving TM3, TM4, TM5, TM6, and TM7. This allosteric coupling influences the $Ca^{2+}$-binding sites' geometry, possibly reducing SERCA's $Ca^{2+}$ binding affinity. In contrast, the analysis of the

**Table 2.** Pairwise inter- and intramolecular hydrogen bond summary for PLN/SERCA REMD PCA clusters.
Parentheses report the average number hydrogen bonds to a charged residue of SERCA or PLN (bold font) observed over all frames assigned to the respective clusters. Hydrogen bonds were defined with a donor -acceptor distance less than 3 Å and a donor-H-acceptor angle less than 20°.

| Cluster | Average hydrogen bonds per frame | | | | | | |
|---|---|---|---|---|---|---|---|
| | E2 | K3 | R9 | R13 | R14 | S16 | E19 |
| 1 | R556 (0.10)<br>K365 (0.08)<br>R638 (0.02) | - | E606 (0.05) | **E19 (0.01)** | E606 (0.01) | - | K328 (0.08)<br>**R13 (0.01)** |
| 2 | R556 (0.10)<br>K365 (0.04) | E644 (0.02)<br>D557 (0.02) | D557 (0.01) | E606 (0.05)<br>**E19 (0.04)** | E606 (0.02) | - | K328 (0.38)<br>**R13 (0.04)** |
| 3 | K365 (0.11)<br>R556 (0.09)<br>R638 (0.01) | D399 (0.01) | E606 (0.02) | - | E606 (0.06) | - | K328 (0.15)<br>R325 (0.04)<br>K329 (0.03) |
| 4 | K397 (0.14)<br>K365 (0.11)<br>R638 (0.05) | D399 (0.24) | D557 (0.10)<br>E392 (0.02)<br>E606 (0.01) | D616 (0.06) | - | - | - |

**Table 3.** Pairwise inter- and intramolecular hydrogen bond summary for pSer16-PLN/SERCA REMD PCA clusters. Hydrogen bonds were measured according to *Table 2*.

| Cluster | Average hydrogen bonds per frame | | | | | | |
|---|---|---|---|---|---|---|---|
| | **E2** | **K3** | **R9** | **R13** | **R14** | **S16** | **E19** |
| 1 | K365 (0.04)<br>R556 (0.03)<br>K397 (0.02) | D557 (0.01) | D557 (0.14)<br>**S16 (0.13)**<br>E392 (0.02) | **S16 (0.59)**<br>E606 (0.02) | **S16 (0.72)**<br>E606 (0.04) | **R14 (0.72)**<br>**R13 (0.59)**<br>R604 (0.54)<br>K605 (0.30)<br>**R9 (0.13)** | K328 (0.07)<br>K329 (0.02) |
| 2 | K365 (0.07)<br>R556 (0.03) | D399 (0.01) | **S16 (0.55)**<br>E606 (0.03) | **S16 (0.21)**<br>E19 (0.06)<br>E748 (0.02)<br>D616 (0.01) | **S16 (0.81)**<br>E606 (0.06) | **R14 (0.81)**<br>**R9 (0.55)**<br>**R13 (0.21)**<br>K605 (0.16)<br>R604 (0.01) | K328 (0.10)<br>**R13 (0.06)** |
| 3 | K365 (0.12)<br>**K3 (0.02)**<br>R556 (0.01) | D399 (0.03)<br>D557 (0.02)<br>**E2 (0.02)** | **S16 (0.82)**<br>E606 (0.05) | **S16 (0.07)**<br>E748 (0.03) | **S16 (0.85)**<br>E606 (0.01) | **R14 (0.85)**<br>**R9 (0.82)**<br>**R13 (0.07)**<br>K605 (0.05) | - |
| 4 | R556 (0.09)<br>K365 (0.06)<br>R560 (0.02)<br>R638 (0.01)<br>K400 (0.01) | D557 (0.03) | **S16 (0.28)**<br>E606 (0.05) | **S16 (0.57)**<br>E19 (0.03) | **S16 (0.86)**<br>E606 (0.02) | **R14 (0.86)**<br>**R13 (0.57)**<br>**R9 (0.28)** | K328 (0.09)<br>**R13 (0.03)**<br>K329 (0.02) |
| 5 | R556 (0.07)<br>K365 (0.05)<br>**K3 (0.03)**<br>K400 (0.03)<br>R638 (0.03) | 399 (0.04)<br>E2 (0.03) | **S16 (0.16)**<br>E606 (0.05) | **S16 (0.77)** | **S16 (0.77)** | **R13 (0.77)**<br>**R14 (0.77)**<br>**R9 (0.16)**<br>K328 (0.03)<br>K329 (0.02) | R325 (0.19)<br>K328 (0.13)<br>K464 (0.02) |
| 6 | R556 (0.06)<br>**K3 (0.04)** | E2 (0.04) | D399 (0.08)<br>E394 (0.05) | **S16 (0.84)** | **S16 (1.14)**<br>E392 (0.25) | **R14 (1.14)**<br>R460 (0.89)<br>**R13 (0.84)**<br>K464 (0.11)<br>K329 (0.07)<br>R325 (0.01) | K328 (0.21)<br>K329 (0.04)<br>K464 (0.03) |
| 7 | K365 (0.14) | - | E19 (0.03)<br>D616 (0.02) | E748 (0.54)<br>E606 (0.03) | **S16 (0.36)** | R325 (1.18)<br>**R14 (0.36)**<br>K329 (0.31)<br>K328 (0.25) | **R9 (0.03)** |
| 8 | K397 (0.10)<br>K400 (0.03) | E644 (0.01) | E394 (0.23) | **S16 (0.67)** | **S16 (1.22)**<br>E392 (0.65) | **R14 (1.22)**<br>**R13 (0.67)**<br>R460 (1.35) | R325 (0.45)<br>K328 (0.44)<br>K329 (0.37) |

trajectories of the SERCA/pPLN complex displays only correlated motions between the TM of pPLN and the most proximal SERCA helices, with only a sparse network of correlations involving TM4, TM5, TM6, and TM8. Phosphorylation of PLN at Ser16 increases the electrostatic interactions with the cytoplasmic domain of SERCA (*R* to *B* state transition, i.e. disorder to order) (*Gustavsson et al., 2013*), and simultaneously weakens intramembrane protein-protein interactions to uncouple the dynamic transitions of PLN from SERCA. The latter removes the structural hindrance of PLN's TM domain and augments $Ca^{2+}$ transport (*Figure 4E,F*).

## Effects of $Ca^{2+}$ ion binding to SERCA on PLN's topology

To assess the effects of $Ca^{2+}$, we performed SLF experiments on SERCA/PLN and SERCA/pPLN complexes in the *E*1 state (*Figure 5A,B*). The addition of $Ca^{2+}$ to the SERCA/PLN complex did not cause significant changes to the PLN topology ($\theta = 32.9 \pm 1.4°$ and $\rho_{L31} = 199 \pm 4°$; p=0.29 and p=0.0025 compared to the *E*2 complex). However, the reappearance (i.e. sharpening) of several resonances in the spectra (e.g. Phe32, Phe35/Leu42, and Ala36) indicates a rearrangement of the binding interface between the two proteins similar to phosphorylation's effect on the *E*2 complex. A small topological change, however, was observed for the SERCA/pPLN complex, for which $Ca^{2+}$-binding induced a decrease of both tilt and rotational angles of $1.8 \pm 1.3°$ and $10 \pm 7°$, respectively

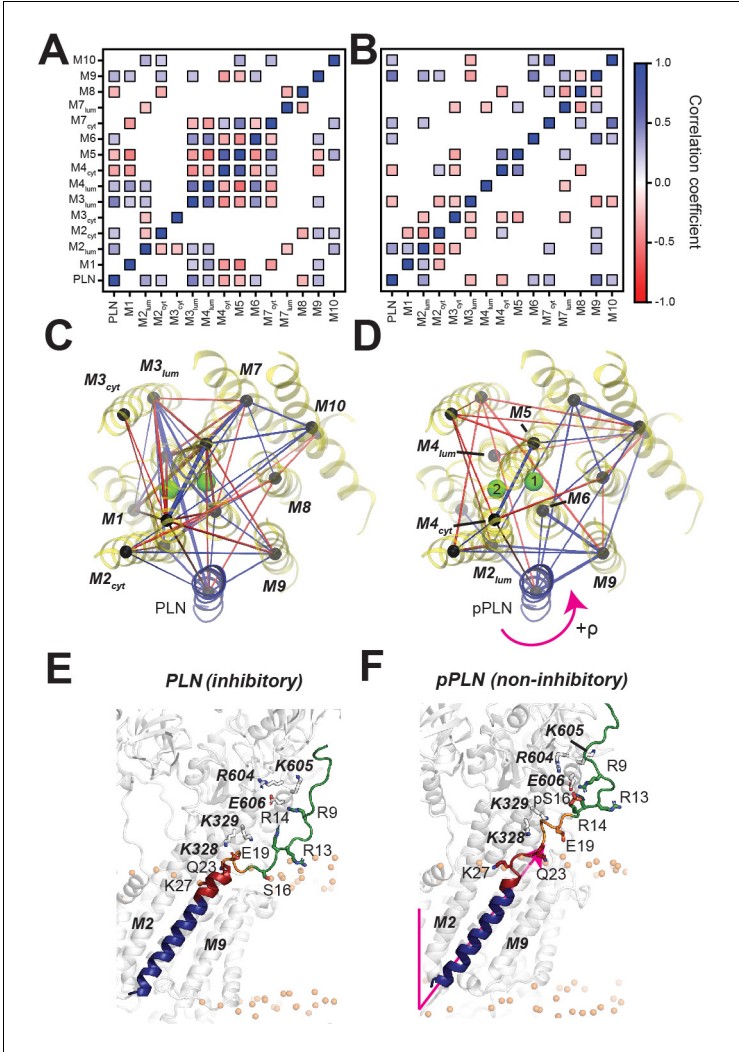

**Figure 4.** PLN topological transitions are allosterically coupled to SERCA's Ca²⁺ binding sites. (A, B) Correlation maps of motions between the TM topology of PLN (A) or pPLN (B) with the topology of the 10 TM domains of SERCA. (C, D) Corresponding spider plots showing the density of correlations are displayed below. Green spheres mark positions of the calcium-binding sites. (E, F) Snapshots of the SERCA/PLN (E) and SERCA/pPLN (F) complexes highlighting the transient interactions with the cytoplasmic region and loosened interactions with the TM region of SERCA.

($\theta = 28.6 \pm 0.7°$ and $\rho_{L31} = 187 \pm 6°$, p=1.9 × 10⁻⁶ and p=4.2 × 10⁻⁷ compared to the $E2$ complex). Due to the lack of X-ray structures, we were unable to carry out dynamic modeling of these complexes. However, these experimental results agree with our $E2$-SERCA models suggesting that a loss of inhibition, either from phosphorylation or Ca²⁺ binding, does not require an extensive structural and topological reconfiguration of PLN's domain II or complete dissociation of the complex.

## Discussion

OS-ssNMR spectroscopy revealed that the TM helix of PLN undergoes a topological equilibrium that is shifted upon phosphorylation, providing direct evidence of the allosteric coupling between the outer membrane regulatory and TM inhibitory regions. Our dynamic modeling of the SERCA/PLN complexes using experimental restraints shows that the structural disorder of the juxtamembrane domain Ib following Ser16 phosphorylation of PLN signals a slight topological change in the TM region that is sufficient to relieve its inhibitory function. This event involves allosteric effects between the inhibitory interactions of domain II of PLN and SERCA's core helices harboring the Ca²⁺ binding

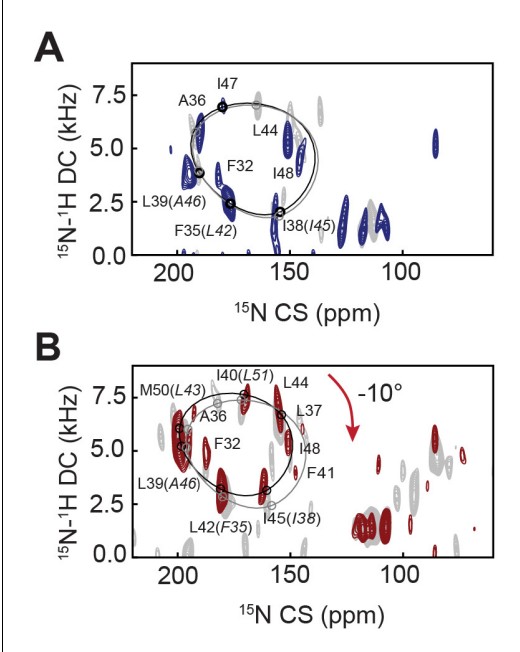

**Figure 5.** Effects of Ca$^{2+}$ binding to SERCA on the topology of PLN and pPLN. (**A, B**) 2D [$^{15}$N-$^1$H] SE-SAMPI4 spectrum of PLN (**A**) and pPLN (**B**) bound to SERCA in the *E*1 form reconstituted into aligned lipid bicelles. PISA wheels for an ideal are superimposed. Equivalent spectra of the *E*2 form complexes are shown in gray.

sites. In addition to the localized disruption of PLN domain Ib interactions with SERCA, phosphorylation and Ca$^{2+}$-binding signal a collective switch of PLN's TM domain from an inhibitory to a non-inhibitory topology. The tilt angle reductions of PLN accompanying the relief of inhibition are easily identifiable from the OS-ssNMR spectra, while rotations are more subtle; nonetheless, these topological changes are sufficient to disrupt critical inhibitory interactions.

Recent X-ray investigations and extensive computational studies have showed that SERCA undergoes significant rocking motions throughout its enzymatic cycle (*Norimatsu et al., 2017*; *Das et al., 2017*; *Rui et al., 2018*). These conformational transitions analyzed in the absence of PLN are highly concerted and cooperative, that is, the dynamics of the cytoplasmic headpiece of SERCA correlates with its TM domains (*Das et al., 2017*; *Rui et al., 2018*). PLN (and other regulins) wedges into the ATPase's binding groove and correlates with the topological changes of SERCA's TM domains or interferes with its rocking motions, leading to uncoupling of ATP hydrolysis and Ca$^{2+}$ transport. PLN experiences significant changes in tilt angle (ranging from 28.6° to 37.5°), depending on both the conformational state of PLN and enzymatic state of SERCA. Therefore, these dynamic and topological transitions provide the mechanism to modulate TM protein-protein interactions, which can

be tuned by posttranslational phosphorylation, O-glycosylation (*Yokoe et al., 2010*), and binding of ancillary proteins (*Kranias and Hajjar, 2017*; *Menzel et al., 2020*).

The structural and dynamics changes of PLN, detected by OS-ssNMR, resolve an ongoing controversy about the *subunit* vs. *dissociative* models proposed for SERCA regulation (*MacLennan and Kranias, 2003*; *Dong and Thomas, 2014*; *Bidwell et al., 2011*; *Mueller et al., 2004*; *Martin et al., 2018*). The latter model speculates that the reversal of the inhibitory function of PLN is due to a complete dissociation of this regulin from the ATPase; but this is not supported by spectroscopic evidence either in vitro or in cell (*Karim et al., 2006*; *Gustavsson et al., 2013*; *Dong and Thomas, 2014*; *Bidwell et al., 2011*; *Martin et al., 2018*). On the other hand, the subunit model agrees well with all spectroscopic measurements, but it does not explain the reversal of inhibition caused by phosphorylation or the elevation of Ca$^{2+}$ concentration. Our ssNMR-driven dynamics calculations clearly show that topological and structural changes modify the interactions at the interface and are propagated to the distal Ca$^{2+}$-binding sites.

*Figure 6* summarizes our proposed mechanistic model for allosteric control of SERCA by PLN's topological changes. We previously showed that PLN's cytoplasmic domain undergoes a three-state equilibrium (*T*, *R*, and *B*) in which the *T* and *R* states are inhibitory, while the *B* state is non-inhibitory. (*Gustavsson et al., 2013*) Our new data show that, when bound to the *E*2-SERCA state, the TM region of PLN (domains Ib and II) remains locked into the ATPase's binding groove. An increase of Ca$^{2+}$ concentration drives SERCA into the *E*1 state and reconfigures the intramembrane binding interface augmenting Ca$^{2+}$ transport without significant topological changes to PLN. On the other hand, detectable topological changes occur upon PLN's phosphorylation both at low and high Ca$^{2+}$ concentrations, shifting the equilibrium toward the non-inhibitory *B* state of PLN (*Masterson et al., 2011*) and transmitting changes across the SERCA/PLN interface that increase Ca$^{2+}$ transport. In this framework, it is possible to explain how single-site disease mutations or deletion in domains Ia and

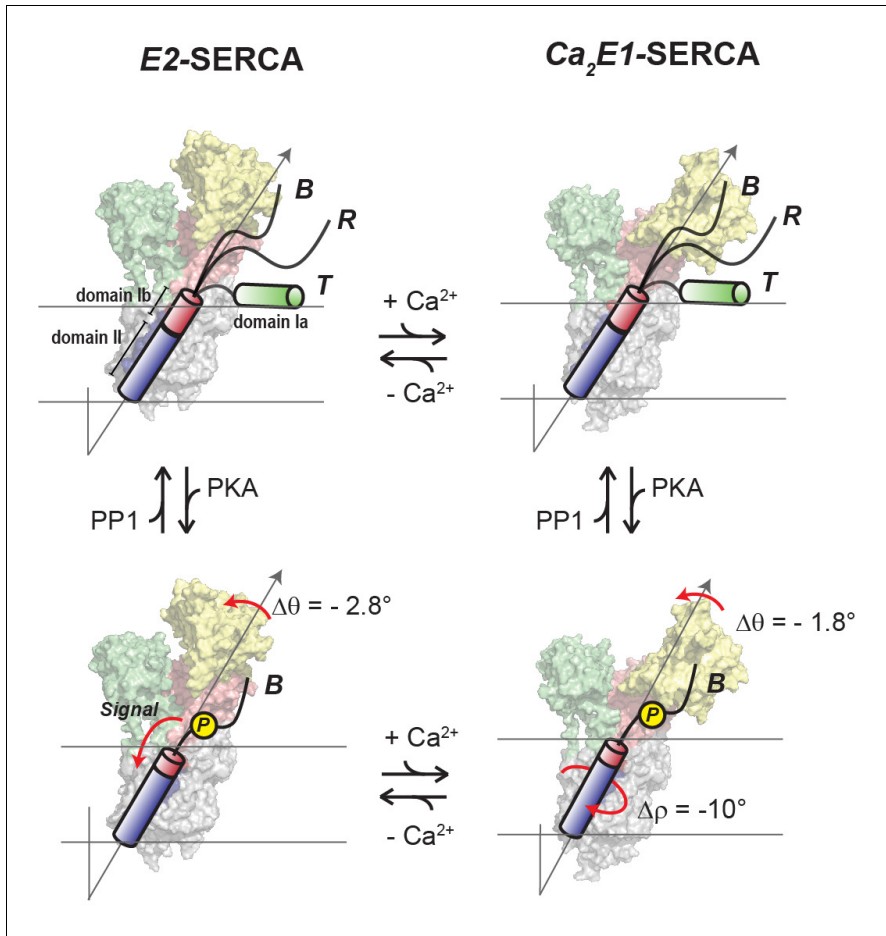

**Figure 6.** Regulatory model of SERCA by PLN's phosphorylation and $Ca^{2+}$. PLN's cytoplasmic domain exists in equilibrium between three distinct populations in the presence of SERCA. The increase of $Ca^{2+}$ ions causes significant shifts in SERCA's conformation toward the $E1$ state, while the topology of PLN is only slightly affected (top equilibrium). Phosphorylation of PLN at Ser16 signals more extensive topological changes with a reconfiguration of SERCA/PLN TM interactions both at low and high $Ca^{2+}$ concentrations (bottom equilibrium), augmenting $Ca^{2+}$ transport. Note that the $T$ and $R$ populations are not represented for clarity.

Ib may lead to perturbations of the protein-protein electrostatic network of interactions, resulting in dysfunctional $Ca^{2+}$ transport (*Kimura et al., 1998*; *Fujii et al., 1989*; *Toyofuku et al., 1994*).

In conclusion, the structural dynamics and topological allostery identified for PLN may explain how bitopic miniproteins, despite their simple architecture, can fulfill diverse regulatory roles and how posttranslational modification at cytoplasmic sites may constitute switches for signal transduction across cellular membranes operated by single or multiple transmembrane domains (*Bocharov et al., 2017*). Several mini-membrane proteins regulate membrane-embedded enzymes or receptors (*Andrews and Rothnagel, 2014*). In the heart, phospholemman (*Presti et al., 1985*; *Bers et al., 2006*; *Teriete et al., 2009*), a member of the FXYD family, regulates the $Na^+/K^+$-ATPase interacting via its transmembrane domain, with its regulatory interactions modulated by protein kinases A and C. Several regulins have also been recently found to control SERCA's isoforms in other tissues (*Anderson et al., 2016*) and share similar topologies to PLN. They all bind at distal locations from the active sites (e.g. ATP or ion channels) of enzymes, revealing possible hotspots for allosteric control by small molecules. Therefore, the characterization of the topological allosteric control of SERCA by PLN represents a first step in understanding how and why evolution has preserved these small polypeptides as a means to regulate the function of ATPases (*Bers et al., 2006*; *Singh et al., 2019*) or other membrane transporters (*Makarewich, 2020*).

# Materials and methods

## Key resources table

| Reagent type (species) or resource | Designation | Source or reference | Identifiers | Additional information |
|---|---|---|---|---|
| Strain, strain background (*Escherichia coli*) | BL21-Codon Plus (DE3)-RP | Agilent | 230255 | Chemically competent cells for protein expression |
| Recombinant DNA reagent | pMAL c2X PLN[AFA] (plasmid) | *Buck et al., 2003* | | MBP fusion with monomeric mutant of rabbit PLN (AFA; C36A, C41F, C46A) |
| Peptide, recombinant protein | PLN[AFA] | This work | NCBI NP_00 1076090.1 | Purified monomeric mutant of rabbit PLN (C36A, C41F, C46A) protein |
| Biological sample (*Oryctolagus cuniculus*) | SERCA1a | This work | NCBI NP_001 082787.1 | Protein extracted and purified from rabbit skeletal muscle |
| Chemical compound, drug | 4-(Trifluoromethyl) benzyl bromide | Sigma-Aldrich | 290564 | Precursor for TMFB-MTS synthesis |
| Chemical compound, drug | Sodium methanethio sulfonate | Sigma-Aldrich | 684538 | Precursor for TMFB-MTS synthesis |
| Chemical compound, drug | 1,2-Dimyristoyl-sn-glycero-3-phosphocholine (DMPC) | Avanti Polar Lipids | 850345 | Bicelle long chain lipid |
| Chemical compound, drug | 1-Palmitoyl-2-oleoyl-glycero-3-phosphocholine (POPC) | Avanti Polar Lipids | 850457 | Bicelle long chain lipid |
| Chemical compound, drug | 1,2-Dimyristoyl-sn-glycero-3-phosphoethanol amine-N-diethylenetriamine pentaacetic acid (ammonium salt) (PE-DTPA) | Avanti Polar Lipids | 790535 | Bicelle long chain lipid for chelating $Yb^{3+}$ |
| Chemical compound, drug | 1,2-Dihexanoyl-sn-glycero-3-phosphocholine (DHPC) | Avanti Polar Lipids | 850305 | Bicelle short chain lipid |
| Chemical compound, drug | Octaethylene Glycol Monododecyl Ether (C12E8) | Anatrace | O330 | Detergent for SERCA1a extraction/purification and reconstitution |
| Chemical compound, drug | Ytterbium(III) chloride hexahydrate | Sigma-Aldrich | 204870 | For flipping bicelles |
| Software, algorithm | NMRFAM-SPARKY | *Lee et al., 2015* | http://pine.nmrfam. wisc.edu/download_ packages.html | Analysis of NMR spectra |
| Software, algorithm | PISA-SPARKY | *Weber et al., 2020a* | http://veglia.chem. umn.edu/software-downloads | Model fitting of SE-SAMPI4 spectra |

*Continued on next page*

*Continued*

| Reagent type (species) or resource | Designation | Source or reference | Identifiers | Additional information |
|---|---|---|---|---|
| Software, algorithm | NMRPipe | *Delaglio et al., 1995* | https://www.ibbr.umd.edu/nmrpipe/ | NMR time-domain processing |
| Software, algorithm | Nmrglue | *Helmus and Jaroniec, 2013* | https://www.nmrglue.com/ | NMR spectra plotting, visualization |
| Software, algorithm | MD2SLF | *Weber and Veglia, 2020* | http://veglia.chem.umn.edu/software-downloads | MD-based prediction of SE-SAMPI4 spectra |
| Software, algorithm | PyMOL 2.3.0 | Schrödinger | https://pymol.org/2/ | Molecular visualization |
| Software, algorithm | VMD 1.9.3 | *Humphrey et al., 1996* | http://www.ks.uiuc.edu/Research/vmd/ | Analysis of MD trajectories |
| Software, algorithm | CHARMM-GUI Membrane Builder | *Wu et al., 2014* | http://www.charmm-gui.org/ | Building of unbiased MD simulations |
| Software, algorithm | MODELLER | *Webb and Sali, 2016* | https://salilab.org/modeller/ | Missing loop construction in models |
| Software, algorithm | AMBER18 and AMBER Tools18 | *Case et al., 2018* | https://ambermd.org/ | Unbiased simulations of PLN |
| Software, algorithm | CPPTraj (included in AMBERTools18) | *Roe and Cheatham, 2013* | https://ambermd.org/ | Principal component analysis |
| Software, algorithm | GROMACS 4.6.7 | *Abraham et al., 2015* | http://www.gromacs.org/ | RAOR-MD and analysis |
| Software, algorithm | RAOR-MD plugin for GROMACS 4.6.7 | *De Simone et al., 2014* | https://github.com/maximosanz/modelSSNMR *Sanz-Hernández, 2021* (copy archived at swh:1:rev:c930a2995a0f2af0e02661d4816fa64cfd8bfe38) | Plugin for implementing |

## Resource availability

### Lead contact

Further information and requests for resources and reagents should be directed to and will be fulfilled by the lead contact, Gianluigi Veglia (vegli001@umn.edu).

## Materials availability

Expression plasmids used in this study are available on request.

## Experimental models and subject details

All experiments were caried out in vitro using PLN obtained in this work by recombinant expression. Recombinant PLN was expressed in *E. coli* BL21 (DE3) cells grown in minimal media required for uniform or selective $^{15}$N labeling. SERCA1a was purified from crude ER stored at −80°C. The crude ER was prepared from skeletal muscle harvested from New Zealand white rabbits immediately following euthanasia (approved IACUC Protocol 1805-35910A).

## Methods details

### Expression, purification, and phosphorylation of PLN

The monomeric cysteine-null mutant of PLN (Cys36Ala, Cys41Phe, and Cys46Ala) was expressed uniformly $^{15}$N-labeled as a soluble fusion with maltose binding protein (MBP) as reported previously (*Buck et al., 2003*), with minor modifications. Freshly transformed *E. coli* CodonPlus (DE3)-RP cells (Agilent) were used to inoculate overnight LB cultures, which were subsequently centrifuged and resuspended into M9 minimal media ($^{15}$NH$_4$Cl as the sole nitrogen source) at an OD$_{600}$ of ~0.7. Cultures were grown at 30°C to OD$_{600}$ 1.0 then induced with 1 mM IPTG over 20 hr to a final OD$_{600}$ of 5. Cells (~6 g/L of M9) were stored at $-20$°C. Selectively $^{15}$N-labeled PLN was expressed from M9 media (free of NH$_4$Cl) with 125 mg/L of the respective $^{15}$N-amino acid, 300 mg/L of non-scrambling and 450 mg/L scrambling-prone $^{14}$N-amino acids (*Lacabanne et al., 2018*). Reverse-labeled PLN was expressed in M9 minimal media ($^{15}$NH$_4$Cl) with 1 g/L of the respective $^{14}$N-labeled amino acid. Induction times for selective and reverse labeling growths were reduced to 3–4 hr to reduce scrambling.

For purification, cells were homogenized (Sorvall Omni Mixer) and lysed by sonication in 200 mL lysis buffer (20 mM sodium phosphate, 120 mM NaCl, 2 mM DTT, 1 mM EDTA, 0.1 mg/mL lysozyme, 0.5% glycerol, 0.5% Tween 20, and protease inhibitors, pH 7.3). The lysate was centrifuged (17,500 rpm, JA25.50 rotor, 4°C, 40 min) and supernatant loaded onto 30 mL bed volume of amylose resin. The resin was washed with buffer (20 mM sodium phosphate, 120 mM NaCl, pH 7.3) and eluted into 100 mL buffer including 50 mM maltose. Elution volumes were concentrated to ~50 mL and dialyzed overnight against 3 L of cleavage buffer (50 mM Tris-HCl, 2 mM β-mercaptoethanol, pH 7.3). All purification steps were done at 4°C and yielded up to 120 mg of fusion protein from 1 L M9 media.

To phosphorylate PLN at Ser16 (pPLN), the MBP fusion was dialyzed into 30 mM Tris-HCl, pH 7.5, followed by addition of 11x reaction buffer to reach 50 mM Tris-HCl, 10 mM MgCl$_2$, 0.05 mM PMSF, 1 mM NaN$_3$, 1 mM (EDTA). The catalytic subunit of protein kinase A (PKA) was added at 1:1000 ratio to fusion protein, with 2 mM DTT, and the reaction started by addition of 2 mM ATP and incubation at 30°C for 3 hr with gentle agitation.

MBP-PLN, either phosphorylated or non-phosphorylated, was cleaved with TEV protease and 2 mM DTT for 3 hr at 30°C to liberate insoluble PLN or pPLN (*Figure 7A*), which was pelleted by centrifugation and dissolved into 10% SDS and 50 mM DTT at approximately 10 mg/mL then stored at $-20$°C. PLN was further purified by HPLC using a Vydac 214TP10154 C4 column heated at 60°C and eluted using H$_2$O/0.1% trifluoroacetic acid (TFA) and a linear gradient of isopropanol/0.1% TFA from 10% to 40% over 10 mins then to 80% over 50 min (2 mL/min flow rate). The protein was lyophilized. Complete phosphorylation of PLN was confirmed by MALDI-MS (*Figure 7B*). The inhibitory activity of PLN against SERCA, and relieved inhibition of pPLN, in the DMPC/POPC (4:1) lipid bilayer composition used for NMR studies, was confirmed by a coupled enzyme assay (*Reddy et al., 2003*; *Figure 7C*).

## Preparation of oriented bicelle samples

Long-chain lipids DMPC (37.0 mg), POPC (10.4 mg) and PE-DTPA (0.9 mg; that is, 79.25:19.75:1.0 molar ratio) were aliquoted together from chloroform stocks (Avanti Polar Lipids), dried to a film with N$_2$ and residual solvent removed under high vacuum. The film was resuspended into 1 mL ddH$_2$O, freeze-thawed three times between liquid N$_2$ and a 40°C water bath then lyophilized. DHPC (7.7 mg; 1:4 molar ratio, or $q = 4$, to long chain lipids) was prepared separately from a chloroform stock, dried and lyophilized from ddH$_2$O.

For bicelles containing only PLN, DHPC was dissolved into 250 µL sample buffer (20 mM HEPES, 100 mM KCl, 1 mM NaN$_3$, 2.5% glycerol, pH 7.0), then used to solubilize PLN (2.5 mg) by vortex. Separately, long-chain lipids were suspended into 250 µL of buffer. PLN in DHPC and long-chain lipids, both pre-chilled in ice, were combined and vortexed while allowing the sample to reach room temperature, then placed back on ice. The process was repeated at least three times to fully solubilize long-chain lipids, which produced a completely transparent liquid at cold temperature (micelle phase) and transparent solid gel at room temperature (bicelle phase). The sample was placed on ice, brought to pH 4.2 with KOH, then concentrated to ~180 µL using a 0.5 mL 10 kDa MWCO centrifugal filter (Amicon) at 4°C. The solubility of PLN and pPLN was significantly diminished at higher pH.

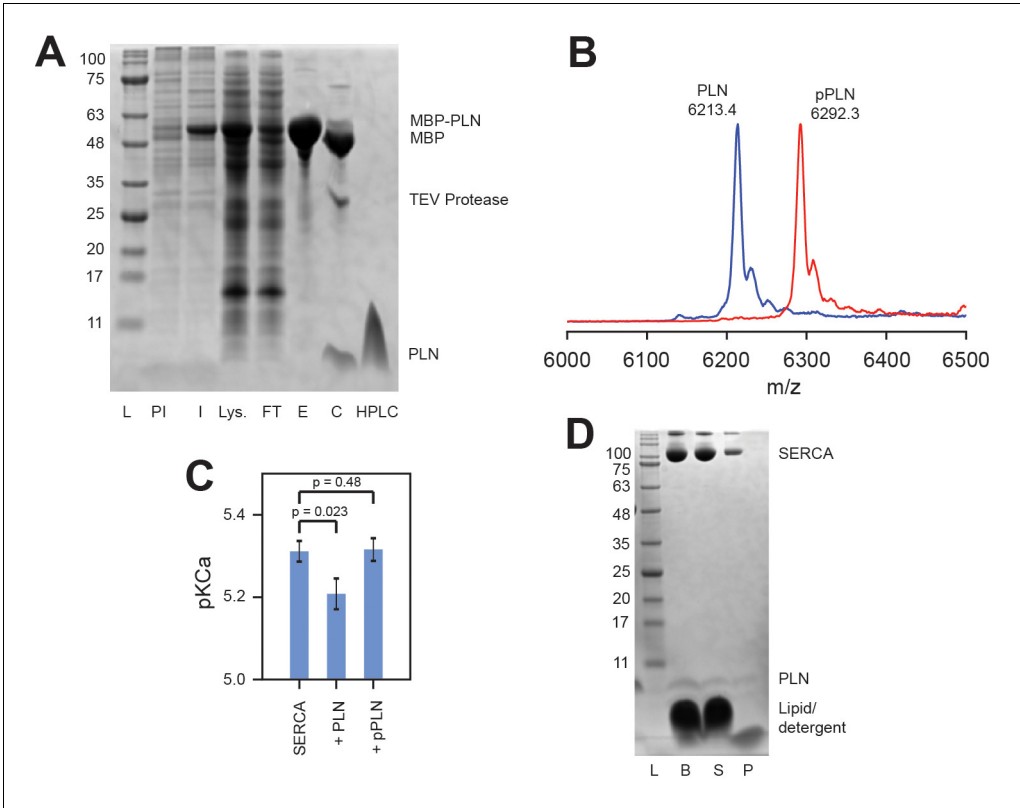

**Figure 7.** Activity and confirmation of PLN. (**A**) SDS-PAGE gel of the expression and purification of PLN as an MBP-fusion: ladder (L), pre-induced (PI) and induced (I) expression, lysate supernatant (Lys.), purified MBP-PLN eluted from amylose resin (E), MBP-PLN after cleavage with TEV protease (C), and HPLC-purified PLN. (**B**) MALDI-MS spectra of HPLC-purified $^{15}$N PLN and pPLN. (**C**) Coupled-enzyme ATPase activity assays of SERCA reconstituted into DMPC/POPC (4:1; lipid-to-protein ratio of 700:1) liposomes with a fivefold excess of either PLN or pPLN. Assays consisted of 50 mM HEPES, 100 mM KCl, 5 mM MgCl$_2$, 0.2 mM NADH, 0.5 mM PEP, 10 U/mL pyruvate kinase, 10 U/mL lactate dehydrogenase, and 7 µM calcium ionophore A23187 at pH 7.0. ATPase activity was measured by the rate of reduction in NADH absorbance at 340 nm at EGTA-buffered calcium concentrations between $10^{-7}$ and $10^{-4}$ M, initiated by addition of 5 mM ATP. Activity as a function of calcium concentration was fitted to a Hill function to extract the pKCa values (calcium concentration at half V$_{max}$) shown. Errors indicate the standard deviation from three replicate measurements. Statistical significance was determined by unpaired *t*-test (Microsoft Excel 365 Version 2002). (**D**) SDS-PAGE gel of SERCA and PLN reconstituted into bicelles for OS-ssNMR: bicelle sample freshly solubilized by DHPC (B), supernatant after centrifugation (S) and pellet formed from a small non-solubilized component (P).

The sample was doped with 0.8 µL of 1 M YbCl$_3$, corrected back to pH 4.2 with KOH, then loaded into a 5 mm flat bottom sample cell (New Era).

For bicelles containing the SERCA/PLN complexes, SERCA1a was purified from rabbit skeletal muscle as previously described (*Stokes and Green, 1990*). SERCA was eluted from Reactive Red affinity resin at ~0.5 mg/mL in SERCA, 0.1% C$_{12}$E$_8$, 1 mM CaCl$_2$, 1 mM MgCl$_2$, 20 mM MOPS, 20% glycerol, 8 mM ADP, 0.25 mM DTT, pH 7.00 and stored at −80°C. Protein concentration was determined by Pierce BCA Assay (Thermo Scientific) and activity confirmed by coupled-enzyme assays (*Reddy et al., 2003*). Immediately prior to use, SERCA (4 mg) was thawed at 4°C and combined with long-chain lipids (prepared as above, except with 1.5% molar PE-DTPA chelating lipid) solubilized into 1 mL of 4% C$_{12}$E$_8$ in sample buffer. The mixture was diluted to ~30 mL with sample buffer and stirred at 4°C for 30 min prior removal of C$_{12}$E$_8$ by adding 4 g of Bio-Beads SM-2 (Bio-Rad) in stages of 0.5, 0.5, 1, and 2 g with 15 min stirring between additions. Stirring continued overnight at 4°C. Bio-Beads were removed by a 25G syringe and the cloudy suspension of proteoliposomes centrifuged at (12,000 rpm, JA25.50 rotor, 4°C, 30 min). The pellet was resuspended into ~40 mL sample buffer and centrifuged once more to wash out residual elution buffer. The final pellet was

resuspended with 250 µL sample buffer and fully solubilized by adding a 250 µL mixture of PLN, or pPLN (0.23 mg), in DHPC (adjusted to pH 7.0) with several cooling/heating cycles under vortex. The bicelle mixture (~1 mL) was centrifuged (13,400 rpm, Eppendorf F45-12-11 rotor, 4°C, 30 s) to remove insoluble debris and the supernatant concentrated to ~200 µL using a 0.5 mL 10 kDa MWCO centrifugal filter (Amicon) at 4°C. The sample was doped with 1.6 µL of 1 M $YbCl_3$ in four stages, correcting pH back to 7.0 with KOH at each addition. Sample buffer included 20 mM HEPES, 100 mM KCl, 1 mM $NaN_3$, 5 mM $MgCl_2$, 2 mM DTT, 2.5% glycerol, pH 7.0 with 4 mM EGTA or 5 mM $CaCl_2$ to stabilize $E2$ or $E1$ states, respectively. SDS-PAGE confirmed co-reconstitution of SERCA and PLN in the bicelles (*Figure 7D*).

## Synthesis of TFMB and tagging of SERCA

Trifluoromethylbenzyl (TFMB)-methanethiosulfonate (MTS) was synthesized analogously to our method previously reported for synthesizing a $^{13}C$-ethylmethanethiosulfonate reagent (*Vostrikov et al., 2016*; *Weber et al., 2019*) with minor modifications. Briefly, 4-(trifluoromethyl) benzyl bromide (5 mmole), MTS (5 mmole) and KI (0.03 mmole) were dissolved into 2 mL of dimethylformamide and stirred under nitrogen for 16 hr at 40°C.

For TFMB tagging, SERCA (8 mg) was thawed and dialyzed into 1 L of $E1$-state sample buffer (as above, but without DDT and including 0.25 mM $C_{12}E_8$) overnight at 4°C. DMPC (25.2 mg) and POPC (7.1 mg), solubilized in 500 µL of 5% $C_{12}E_8$, was then added to dialyzed SERCA. Detergent was removed by stirring with 1 g Bio-Beads SM-2 for 1.5 hr at 4°C and a further 30 min at room temperature. Proteoliposomes were removed from Bio-Beads with a 25G syringe and centrifuged (18,000 rpm, JA25.50 rotor, 4°C, 40 min). The pellet was suspended into 120 µL buffer and solubilized by adding DHPC ($q = 4$ for oriented or $q = 0.5$ for isotropic bicelles). TFMB-MTS (200 mM in DMSO) was added to the bicelles at 5:1 molar excess and incubated at room temperature for 1 hr prior to concentrating to ~250 with a 0.5 mL 10 kDa MWCO centrifugal filter (Amicon) and loading into a Shigemi NMR tube for $^{19}F$ NMR measurement.

$^{19}F$ NMR spectra of TFMB-tagged SERCA were acquired on a solution state Bruker 600 MHz Avance NEO spectrometer equipped with a TCI HCN cryoprobe. 1D single-pulse experiments were acquired using a 90° pulse of 12 µs and recycle delay of 0.4 s. Spectra in isotropic bicelles were acquired with 1 k scans and 4 k scans for oriented bicelles. Spectra were processed using NMRPipe (*Delaglio et al., 1995*).

## Oriented solid-state NMR spectroscopy

All $^{15}N$ spectra were acquired on a Varian VNMRS spectrometer equipped with a low-$E$ static bicelle probe (*Gor'kov et al., 2007*) operating at a $^1H$ frequency of 700 MHz. 1D [$^1H$-$^{15}N$]-cross-polarization (CP)-based experiments used 90° pulse length of 5 µs, or 50 kHz radiofrequency (RF) field, on $^1H$ and $^{15}N$ channels; contact time of 500 µs with a 10% linear ramp on $^1H$ centered at 50 kHz; and an acquisition time of 10 ms under 50 kHz SPINAL64 heteronuclear proton decoupling (*Fung et al., 2000*). For all experiments, $^{15}N$ was set to 166.3 ppm and externally referenced to $^{15}NH_4Cl$ at 39.3 ppm (*Bertani et al., 2014*); and detected using a spectral width of 100 kHz.

2D separated local field (SLF) spectra were collected using a signal-enhanced (SE)-SAMPI4 experiment (*Gopinath and Veglia, 2009*; *Gopinath et al., 2010*; *Nevzorov and Opella, 2007*). The indirect dipolar dimension utilized complex points and a spectral width of 31.25 kHz. The $t_1$ evolution period utilized $^1H$ homonuclear decoupling with an RF field of 50 kHz and 48 µs dwell time, and a phase-switched spin-lock pulses on $^1H$ and $^{15}N$ of 62.5 kHz RF field. The sensitivity enhancement block used a τ delay of 75 µs and three cycles of phase-modulated Lee-Goldberg (PMLG) homonuclear decoupling (*Vinogradov et al., 1999*) with an effective RF field of 80 kHz.

3D SE-SAMPI4-PDSD spectra (*Mote et al., 2011*) were acquired with 15 increments in both indirect dimensions; spectral widths of 31.25 kHz and 10 kHz in the dipolar coupling and indirect $^{15}N$ dimensions, respectively; and 3 s mixing time for $^{15}N$-$^{15}N$ diffusion (*Traaseth et al., 2010*). Total acquisition times for PLN samples were typically 1 hr for a 1D [$^1H$-$^{15}N$] CP spectra (1 k scans), 40 hr for 2D SE-SAMPI4 spectra (1 k scans) and 2 weeks for 3D SE-SAMPI4-PDSD spectra (two experiments added with 0.25 k scans each). For the SERCA/PLN complexes, 2D SE-SAMPI4 spectra were acquired over 4.5 days at 25°C (4 k scans, 15 indirect points) and 1D [$^1H$-$^{15}N$] CP spectra for 4 hr (4 k scans). A recycle delay of 3 s was used for all experiments. All spectra were processed using

NMRPipe (*Delaglio et al., 1995*) and analyzed using NMRFAM-SPARKY (*Lee et al., 2015*) and Nmrglue (*Helmus and Jaroniec, 2013*).

## PISA wheel simulations and fitting

Polar Index Slant Angle (PISA) wheels (*Denny et al., 2001*; *Marassi and Opella, 2000*) were fitted to 2D SE-SAMPI4 spectra using the PISA-SPARKY plugin in the NMRFAM-SPARKY package (*Weber et al., 2020a*). Default parameters were used to describe ideal helix structure and $^{15}$N chemical shift (CS) and $^{15}$N-$^1$H dipolar coupling (DC) tensors. The Cα-N-H bond angle was modified from 116° to 119° to best fit PLN spectra. CSs and DCs were fit by exhaustively searching tilts (θ), rotations ($\rho_{L31}$, i.e. referenced to residue Leu31) and order parameters (S) in increments of 0.1°, 1.0°, and 0.01, respectively, for the lowest RMSD between calculated and experimental values (*Weber et al., 2020a*). The parameter S, which factors scaling due to rigid-body helical fluctuations and imperfect alignment (*Weber et al., 2020b*), was determined as 0.80 ± 0.05 and 0.81 ± 0.03 for PLN and pPLN, respectively, alone in bicelles. Due to the sparsity of peaks assigned for spectra of PLN in complex with SERCA, S was fixed to 0.80 to reduce fitting errors. Errors in tilt θ, $\rho_{L31}$ and S were determined by repeating fitting 20 times with peak positions randomly adjusted at each iteration. Random adjustments were taken from a Gaussian distribution having a standard deviation equal to average FWHM peak linewidths (3 ppm for CS and 0.8 kHz for DC dimensions). The statistical significance of topological comparisons made throughout the text were determined using an unpaired *t*-test (Microsoft Excel 365 Version 2002) on tilt and azimuthal angles from the 20 repeat fits.

## Unrestrained molecular dynamics

A simulation of truncated monomeric PLN (Met20 to Leu52; PDB 2LPF *De Simone et al., 2013*) in 97 DMPC and 32 POPC, 150 mM KCl and 4927 waters was constructed using the CHARMM-GUI webserver (*Wu et al., 2014*; *Jo et al., 2008*) and CHARMM36 forcefield (*Huang and MacKerell, 2013*). Production runs were done using the AMBER18 (*Case et al., 2018*; *Crowley et al., 2009*) package at 10 Å cutoff and 8 Å force-based switching and default configuration files provided by the CHARMM-GUI webserver. For example, using the Langevin thermostat (*Loncharich et al., 1992*) (310 K), Monte Carlo barostat (*Faller and de Pablo, 2002*) (1 bar, semi-isotropic coupling) and the SHAKE algorithm (*Miyamoto and Kollman, 1992*) for constraining hydrogens. The simulation was run for 1 μs. The final 900 ns of trajectory was used to predict $^{15}$N chemical shifts and $^{15}$N-$^1$H dipolar couplings, as previously reported (*Weber and Veglia, 2020*).

## NMR-restrained refinement of the SERCA/PLN complex

The initial conformation of SERCA in the *E*2 state was obtained from the crystal structure of the SERCA/PLN complex (PDB 4Y3U) (*Akin et al., 2013*). Missing loops in the structure of SERCA were introduced using MODELLER software (*Webb and Sali, 2016*). Since the super-inhibitory mutant of PLN used for crystallization harbors native cysteines and four substitutions not present in the monomeric mutant used in NMR experiments, as well as a missing C-terminal helical turn, the interface between SERCA and monomeric PLN was refined in silico by introducing information from cross-linking data. Specifically, the transmembrane section of PLN was docked onto SERCA *in vacuo*, maintaining backbone positional restraints on the pump and restraining the dihedral angles of PLN to retain the helical structure. Docking was guided by cross-linking data (*Toyoshima et al., 2003*; *Chen et al., 2006*; *Chen et al., 2003*), performing short MD runs (500 ps) with an harmonic upper wall potential applied to restrain the distances between cross-linked residues to below 5 Å. 100 such runs were performed starting from different orientations of PLN, and the resulting docked complexes were clustered according to the backbone RMSD of the proteins. The center of the most highly populated cluster was picked as the most representative structure and used to continue the modeling. Based on pairwise contacts between SERCA and PLN residues, the register of helix-helix packing was the same between our refined models and the original crystal structure. The N-terminal segment of PLN, not present in the original structure, was built as a random coil detached from SERCA. Ser16 was modeled both with and without the phosphorylation, generating two different PLN/SERCA complexes.

The structures of the PLN/SERCA and pPLN/SERCA complexes were then embedded in DMPC:POPC bilayers, mimicking the experimental conditions, and solvated with TIP3P water

(*Jorgensen et al., 1983*). To equilibrate protein-lipid systems, we employed the two-step multiscale procedure (*Stansfeld et al., 2015*) in which the systems were equilibrated for 1 μs with positional restraints using the coarse-grained model MARTINI (*Marrink et al., 2007*) and then re-converted to full-atom descriptions using the Backward approach (*Wassenaar et al., 2014*). The full-atomic systems were equilibrated for 50 ns at 300 K and a further 50 ns after releasing the positional restraints. Eight equally spaced structures were extracted from the final 20 ns of sampling. These eight structures were used as starting points for ssNMR-restrained replica-averaged sampling (RAOR-MD). The final equilibrated box (of dimensions 10.8 × 10.8×15.9 nm$^3$) contains 247 DMPC lipids, 82 POPC molecules, 39,017 TIP3P waters and 23 Na$^+$ ions to neutralize the system (~174,000 atoms).

$^{15}$N chemical shift (CS) and $^{15}$N-$^1$H dipolar coupling (DC) restraints were incorporated into the sampling using replica-averaged restrained MD, as previously described (*De Simone et al., 2014*; *Sanz-Hernández et al., 2016*) (code available at https://github.com/maximosanz/modelSSNMR), composed of eight replicas evolving simultaneously. The restraining forces were gradually incorporated during an initial equilibration phase of 20 ns, where the forces were linearly increased to a maximum of 50 J/(mol·ppm$^2$) and 800 J/(mol·kHz$^2$) for CSs and DCs, respectively.

In order to enhance the conformational sampling of the disordered domain Ia of PLN, we implemented a sampling based on annealing cycles, whereby the PLN N-terminus periodically binds and detaches from the SERCA surface. The experimental PRE measurements (*Gustavsson et al., 2013*) were incorporated in order to drive each binding event. At the beginning of each cycle, the domain Ia of PLN is fully detached from SERCA by introducing a lower wall harmonic potential that pushes PLN residues away from SERCA residues present in the cytoplasmic domains of the pump. The lower wall potential pushes PLN residues 0 to 10 at least 50 Å away from Ile140 (SERCA A domain), Thr430 (SERCA N domain), and Cys674 (SERCA P domain). Residues 11–14 were pushed at least 25 Å away from those residues and in addition the interfacial SERCA residues Thr742, Ala327, and Leu119. The detachment occurs gradually, by linearly increasing the force of the lower-wall potential to a maximum of 5 J/(mol · nm$^2$) after 500 ps. During this time, the temperature is also linearly increased to 370 K to enhance the conformational sampling space of the disordered domain Ia. CS and DC force constants were linearly decreased to half their value to avoid instabilities during this high-energy phase of the cycle. This detachment step yields a fully detached PLN domain Ia with no SERCA contacts and enough surrounding free space to explore its unbound disordered conformational space. This step is followed by another 500 ps of sampling at 370 K, where the detached PLN domain Ia is allowed to fluctuate. Subsequently, a binding stage follows, whereby the lower-wall potential is linearly removed over 1 ns of sampling. The PRE-derived distance restraints were gradually incorporated over this time as an upper-wall potential at 30 Å with a maximum force of 5 J/(mol·nm$^2$). During this step the force constants of CS and DC are also restored to their full value, and the temperature is linearly decreased to 300 K. After the binding phase, the PLN N-terminus adopts a SERCA-bound conformation. This step leads onto the sampling phase of the cycle, where the restraint forces and the temperature are kept constant for 2 ns. The structures sampled during these 2 ns are the ones included in the final conformational ensembles of SERCA/PLN. This annealing cycle sampling approach is illustrated in *Figure 8*.

We performed 25 annealing cycles per replica for both PLN and pPLN, resulting in a total simulation time of 0.8 μs for each ensemble. 10,000 equally separated structures (400 ns) in the sampling part of the cycles were extracted to perform the analyses described in the main text. All samplings were performed using a previously described version of GROMACS (*Pronk et al., 2013*), modified to include the CS and DC restraints (*Sanz-Hernández et al., 2016*). The CHARMM36 force field (*Huang and MacKerell, 2013*) was used. Temperature was coupled using the v-rescale algorithm (*Bussi et al., 2007*) and pressure was coupled at 1 bar using the semi-isotropic Berendsen method (*Berendsen et al., 1984*). All simulations were carried out under periodic boundary conditions. The integration timestep was set to 2 fs and the LINCS algorithm was used for constraints (*Hess et al., 1997*). Electrostatic interactions were accounted for using the Particle Mesh Ewald method (*Darden et al., 1993*). VMD (*Humphrey et al., 1996*) was used for contact, hydrogen bond and distance measurements; custom Python scripts for computing helical tilt (θ) and rotation (ρ) angles; and GROMACS energy tool for electrostatic interactions. For the principal component analysis (PCA), both SERCA/PLN and SERCA/pPLN ensembles were reduced to only the backbone atoms, combined, and RMS fit to the overall average coordinates prior to computing the coordinate covariance matrix using CPPTraj (*Roe and Cheatham, 2013*; *Galindo-Murillo et al., 2015*). This ensured that

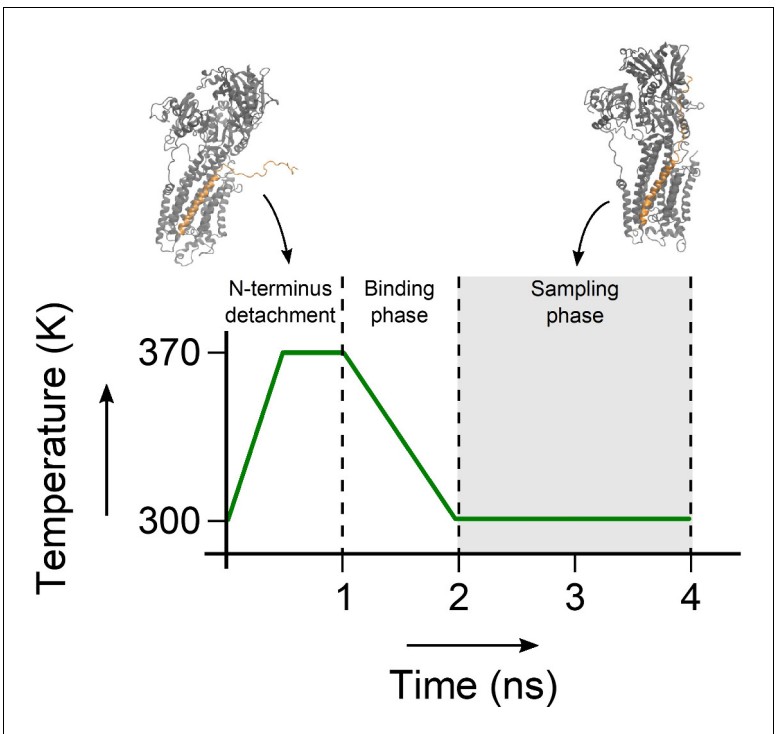

**Figure 8.** Annealing cycle used in the sampling of the SERCA/PLN complex. During the first nanosecond of the simulation, the N-terminus of PLN is detached from SERCA and the temperature is raised to 370 K to randomize its conformation. Then the system is cooled back down to 300 K over one ns, during which time the CS, DC and PRE restraints are re-introduced. The complex is sampled for two ns, after which the cycle restarts.

global rotational and translational movements were removed, and eigenvectors obtained for each ensemble could be directly compared. A PCA analysis was done with PLN (or pPLN) bound to construct videos of the PCA modes, and without PLN for clustering and the projection of X-ray structures illustrated in the main text. Based in the eigenvalues, the first two principal components accounted for 52% the overall motion out of 100 modes calculated from the ensembles. Convergence was assessed by Kullback-Leibler divergence (KLD) of the evolving 2D histogram of the projections (PC1 vs. PC2) computed using data successively truncated from time $t = 0$ to time $t$ compared against the histogram determined using all timepoints:

$$KLD(t) = \sum_{i=0}^{M} p(x_i, t) \cdot \ln\left(\frac{p(x_i, t)}{p(x_i)}\right),$$

where $p(x_i)$ is the final normalized intensity of each bin $i$ and $M$ is the total number of bins ($75^2$). $p(x_i, t)$ is the normalized bin intensity at variable time $t$.

## Acknowledgements

This work was supported by the National Institute of Health grants R01 GM064742 and R01 HL144100 to GV and R01HL139065 and R37AG026160 to DDT and RLC; and European Research Council (CoG - BioDisOrder - 819644) funding to ADS DW was supported by an American Heart Association Postdoctoral Fellowship (19POST34420009). The authors thank Dr. Sanz-Hernández for his initial contribution to the project.

## Additional information

### Funding

| Funder | Grant reference number | Author |
| --- | --- | --- |
| National Institutes of Health | GM064742 | Gianluigi Veglia |
| National Institutes of Health | HL144100 | Gianluigi Veglia |
| National Institutes of Health | HL139065 | David D Thomas |
| National Institutes of Health | AG026160 | David D Thomas |
| European Commission | BioDisOrder - 819644 | Alfonso De Simone |
| American Heart Association | 19POST34420009 | Daniel K Weber |

The funders had no role in study design, data collection and interpretation, or the decision to submit the work for publication.

### Author contributions

Daniel K Weber, Conceptualization, Data curation, Software, Formal analysis, Funding acquisition, Validation, Visualization, Methodology, Writing - original draft, Writing - review and editing; U Venkateswara Reddy, Investigation, Methodology, Writing - review and editing; Songlin Wang, Validation, Investigation, Writing - review and editing; Erik K Larsen, Validation, Investigation, Writing - original draft, Writing - review and editing; Tata Gopinath, Supervision, Validation, Investigation, Methodology, Writing - review and editing; Martin B Gustavsson, Validation, Investigation, Methodology, Writing - review and editing; Razvan L Cornea, David D Thomas, Conceptualization, Writing - original draft, Writing - review and editing; Alfonso De Simone, Conceptualization, Supervision, Validation, Investigation, Visualization, Methodology, Writing - original draft, Writing - review and editing; Gianluigi Veglia, Conceptualization, Resources, Data curation, Formal analysis, Supervision, Funding acquisition, Investigation, Methodology, Writing - original draft, Project administration, Writing - review and editing

### Author ORCIDs

Daniel K Weber ![iD] https://orcid.org/0000-0001-8400-767X
Songlin Wang ![iD] https://orcid.org/0000-0002-7588-7377
Erik K Larsen ![iD] https://orcid.org/0000-0002-7553-212X
David D Thomas ![iD] http://orcid.org/0000-0002-8822-2040
Alfonso De Simone ![iD] http://orcid.org/0000-0001-8789-9546
Gianluigi Veglia ![iD] https://orcid.org/0000-0002-2795-6964

### Decision letter and Author response

Decision letter https://doi.org/10.7554/eLife.66226.sa1
Author response https://doi.org/10.7554/eLife.66226.sa2

## Additional files

### Supplementary files

- Transparent reporting form

### Data availability

Assigned CS-DC correlations from oriented SE-SAMPI4 spectra of non-phosphorylated and phosphorylated monomeric PLN, alone and in complex with SERCA, along raw and processed time-domain data, have been deposited on the Biological Magnetic Resonance Bank with accession codes: Monomeric phospholamban in oriented bicelles; 50719: Mono:meric phosphorylated phospholamban in oriented bicelles; 50720: Phospholamban bound to SERCA in oriented bicelles (calcium-free E2 state); 50721: Phospholamban bound to SERCA in oriented bicelles (calcium-bound E1

state); 50722: Phosphorylated phospholamban bound to SERCA in oriented bicelles (calcium-free E2 state); 50723: Phosphorylated phospholamban bound to SERCA in oriented bicelles (calcium-bound E1 state). RAOR-MD ensembles of PLN and pPLN in complex with SERCA are available in the Data Repository for the University of Minnesota (https://doi.org/10.13020/bkja-jq93).

The following datasets were generated:

| Author(s) | Year | Dataset title | Dataset URL | Database and Identifier |
|---|---|---|---|---|
| Weber DK | 2021 | Ensembles from dynamic refinement of non-phosphorylated and phosphorylated phospholamban-SERCA complexes | https://conservancy.umn.edu/handle/11299/218010 | Data Repository, 218010 |
| Weber DK, Sanz-Hernandez M, Venkateswara Reddy U, Songlin W, Larsen E, Gopinath T, Gustavsson M, Cornea R, Thomas D, De Simone A, Veglia G | 2021 | Monomericphospholamban inoriented bicelles | https://bmrb.io/data_library/summary/?bmrbId=50718 | Biological Magnetic Resonance Data Bank, 10.13018/BMR50718 |
| Weber DK, Sanz-Hernandez M, Venkateswara Reddy U, Songlin W, Larsen E, Gopinath T, Gustavsson M, Cornea R, Thomas D, De Simone A, Veglia G | 2021 | Monomericphosphorylatedphospholamban inoriented bicelles | https://bmrb.io/data_library/summary/?bmrbId=50719 | Biological Magnetic Resonance Data Bank, 10.13018/BMR50719 |
| Weber DK, Sanz-Hernandez M, Venkateswara Reddy U, Songlin W, Larsen E, Gopinath T, Gustavsson M, Cornea R, Thomas D, De Simone A, Veglia G | 2021 | Phospholambanbound to SERCA inoriented bicelles (calcium-free E2 state) | https://bmrb.io/data_library/summary/?bmrbId=50720 | Biological Magnetic Resonance Data Bank, 10.13018/BMR50720 |
| Weber DK, Sanz-Hernandez M, Venkateswara Reddy U, Songlin W, Larsen E, Gopinath T, Gustavsson M, Cornea R, Thomas D, De Simone A, Veglia G | 2021 | Phospholambanbound to SERCA inoriented bicelles (calcium-bound E1 state) | https://bmrb.io/data_library/summary/?bmrbId=50721 | Biological Magnetic Resonance Data Bank, 10.13018/BMR50721 |
| Weber DK, Sanz-Hernandez M, Venkateswara Reddy U, Songlin W, Larsen E, Gopinath T, Gustavsson M, Cornea R, Thomas D, De Simone A, Veglia G | 2021 | Phosphorylatedphospholambanbound to SERCA inoriented bicelles(calcium-free E2state) | https://bmrb.io/data_library/summary/?bmrbId=50722 | Biological Magnetic Resonance Data Bank, 10.13018/BMR50722 |
| Weber DK, Sanz-Hernandez M, Venkateswara Reddy U, Songlin W, Larsen E, Gopinath T, Gustavsson M, | 2021 | Phosphorylatedphospholambanbound to SERCA inoriented bicelles (calcium-bound E1 state) | https://bmrb.io/data_library/summary/?bmrbId=50723 | Biological Magnetic Resonance Data Bank, 10.13018/BMR50723 |

Cornea R, Thomas
D, De Simone A,
Veglia G

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
