## [Decision Letter]

**Acceptance summary:**

Phospholamban, a single transmembrane helix protein, regulates the sarcoplasmic reticulum Ca^2+^-ATPase by binding in the membrane. The work presented here combines new experiments with computer simulations with the aim of arriving at a more definitive answer to the long-standing mechanistic question of how exactly phosphorylation of phospholamban modulates its regulatory behavior. In this manuscript, an allosteric mechanism is presented, which could be of general importance for the whole family of these mini-proteins.

**Decision letter after peer review:**

Thank you for submitting your article "Structural Basis for Allosteric Control of the SERCA-Phospholamban Membrane Complex by Ca^2+^ and Phosphorylation" for consideration by *eLife*. Your article has been reviewed by 3 peer reviewers, one of whom is a member of our Board of Reviewing Editors, and the evaluation has been overseen by Volker Dötsch as the Senior Editor.

The reviewers have discussed their reviews with one another, and the Reviewing Editor has drafted this to help you prepare a revised submission. While the NMR part was seen as very exciting and technically remarkable, the MD simulation was seen more critically. In particular questions about sampling and convergence were raised. Essential revisions are listed below.

Essential revisions:

1. Experimentally it was shown that the type of lipid has a strong influence on the behavior of Phospholamban. Can the authors provide more insight into the protein-lipid interaction from their MD simulation? This is probably in particular important for the conformation of the cytoplasmatic part of Phospholamban.

2. The analysis of the simulations would be strengthened by replacing the principal component analysis with the construction of a Markov State Model. That step in itself would also provide metrics for judging whether sampling has converged sufficiently.

3. The value of the MARTINI simulations is unclear and seeing such simulations distracts from the rigor of the work otherwise.

4. Some concerns about the annealing cycles exist that may affect the rigorousness of the simulation results. A better approach would be more extended biased sampling (such as metadynamics) followed by the application of appropriate unbiasing techniques.

5. There really is not enough information how exactly the simulations were constrained.

---

## [Author Response]

Essential revisions:1. Experimentally it was shown that the type of lipid has a strong influence on the behavior of Phospholamban. Can the authors provide more insight into the protein-lipid interaction from their MD simulation? This is probably in particular important for the conformation of the cytoplasmatic part of Phospholamban.

We agree with this comment. We now describe this point in more detail. As our group and others have found (Gustavsson et al. 2011, J. Mol. Biol.; Gustavsson et al. 2012, BBA), protein-lipid interactions play an important role in the SERCA/PLN complex regulation. Specifically, lipid interactions influence the conformational equilibrium of PLN, affecting the conformation of domain Ia, which toggles between T (inhibitory membrane bound helix) and R (less-inhibitory membrane-detached random coil) states. In our current study, however, we restrained the system using experimental data sampling the membrane-detached R and B (non-inhibitory) states of both the SERCA/PLN and SERCA/pPLN complexes. In fact, the boundary constraints enable domain Ia to sample both free and SERCA-bound states, as mapped out by paramagnetic relaxation enhancements (PRE). These restraints preclude PLN sampling the membrane-bound T state. While most of domain Ia residues do not interact with the membrane, PLN residues Arg13, Arg14, Ser16, and Thr17 are still free to exchange between SERCA-bound and lipid-bound states. For phosphorylated PLN (pPLN), the interactions between domain Ia and lipids are not present, as phosphorylation promotes the B state, in agreement with experimental data (Gustavsson et al. 2013, PNAS). This trend is inverted for domain Ib, which for pPLN detaches from SERCA and binds more strongly to lipids. To clarify this, we have included a new Figure 3A describing a per-residue (PLN) lipid-binding profile (see also lines L305-308 and L314-316).

2. The analysis of the simulations would be strengthened by replacing the principal component analysis with the construction of a Markov State Model. That step in itself would also provide metrics for judging whether sampling has converged sufficiently.

The Markov State Model (MSM) suggested by this reviewer is an excellent method to analyze the conformational landscape of proteins and describe the convergence of the conformational ensembles. However, it cannot be combined with experimental restraints in ensemble-averaging mode as in our study. Even if MSM were to be applicable to our experimentally restrained samplings, the construction of a converged and detailed MSM would require an order-of-magnitude more sampling, which would be prohibitively expensive for a system of this size.

The PCA reveals clear differences for SERCA conformation between the two ensembles. Also, the two principal components (PC1 and PC2) cover 52% of the dynamics described in these samplings and are sufficient to fully resolve the major states identified by X-ray crystallography. Therefore, to address this reviewer’s concern, we have quantitatively assessed the convergence of our calculation and the differences of these conformational landscapes relative to the principal components PC1 and PC2. We have added a new figure to describe the convergence of the system.

Overall, we have included the following points to fully address this reviewer’s concerns:

1. Determined the convergence of the CS and DC observables throughout the annealing cycles (Figure 2—figure supplement 3A, referred to at L219-220)

2. Amended the text to briefly clarify the purpose of the PCA (L224-226).

3. Assessed the convergence for the PCA histograms using Kullback-Leibler divergence (KLD), similar to the method described by Galindo-Murillo et al. 2015, BBA (Figure 2—figure supplement 3B; L234-237).

4. Provided additional details about the KLD analysis and how the PCA was performed, to ensure the principal components were the same between the two ensembles (L717-729).

5. The percent dynamics covered by the first two principal components is now stated L722-723.

6. Provided a summary statement about the potential implications of the divergent SERCA conformational landscapes of the two ensembles, while acknowledging the lack of experimental data to verify this (L250-252).

3. The value of the MARTINI simulations is unclear and seeing such simulations distracts from the rigor of the work otherwise.

We have clarified the value of Martini simulations. This protocol is widely established in the field of membrane protein simulations (L661-662). Denoted as a “two-step” multiscale procedure (Stansfeld et al. 2015, Structure), this protocol starts with a CG simulation to generate stable configuration of the protein in lipid bilayers, followed by the conversion to an atomistic representation and subsequent simulations. In the second step, we employed an approach optimized by the Tieleman lab, slightly modified by the Sansom group to transform from CG to full atomistic configurations (Stansfeld and Sansom 2011, J. Chem. Theory Comput., doi:10.1021/ct100569y).

4. Some concerns about the annealing cycles exist that may affect the rigorousness of the simulation results. A better approach would be more extended biased sampling (such as metadynamics) followed by the application of appropriate unbiasing techniques.

Due to the ensemble averaging of the NMR restraints, metadynamics is not an efficient method, and its convergence becomes virtually impossible with the replica averaging (De Simone et al. 2014, Biophys. J., doi: 10.1016/j.bpj.2014.03.026; Krieger et al. 2014, Biophys. J., doi: 10.1016/j.bpj.2014.03.004). The employed annealing cycles protocol has shown to be effective and convergent on a number of different NMR restraints in replica averaging mode, including RDCs (De Simone et al. 2013, Biochemistry, doi: 10.1021/bi400517b; De Simone et al. 2009, J. Am. Chem. Soc., doi: 10.1021/ja8087295), H/D exchange (De Simone et al. 2011, PNAS, doi: 10.1073/pnas.1112197108), isotropic chemical shifts (Camilloni et al. 2012, J. Am. Chem. Soc., doi: 10.1021/ja210951z; Krieger et al. 2014, Biophys. J., doi: 10.1016/j.bpj.2014.03.004), and finally non-NMR restraints such as Phi values (Gianni et al. 2010, Nat. Struct. Mol. Biol., doi: 10.1038/nsmb.1956).

5. There really is not enough information how exactly the simulations were constrained.

We now provide greater details about how the N-terminus was restrained (L202-211), and a more detailed schematic is now illustrated in Figure 2—figure supplement 1A. Distance distributions of all pairwise boundary restraints have also been supplied (Figure 2—figure supplement 1B).